# 3000 yr-old patterns of mobile pastoralism revealed by multiple isotopes and radiocarbon dating of ancient horses from the Mongolian Altai

Antoine Zazzo[1]*, Maël Le Corre[1,2], Nicolas Lazzerini[1], Charlotte Marchina[3,4], Noost Bayarkhuu[5,6], Vincent Bernard[7], Mathilde Cervel[8], Denis Fiorillo[1], Dominique Joly[9], Michel Lemoine[1], Philippe Telouk[2], François Thil[10], Tsagaan Turbat[5], Vincent Balter[2], Aurélie Coulon[11,12], Sébastien Lepetz[1]

**1** Bioarchéologie, Interactions Sociétés Environnements (BioArch, UMR 7209), Muséum National d'Histoire Naturelle, Sorbonne Université, Centre National de la Recherche Scientifique (CNRS), Paris, France, **2** Laboratoire de Géologie de Lyon, Terre, Planètes, Environnement (LGLTPE, UMR 5276), École Normale Supérieure Lyon, Université Lyon 1, Centre National de la Recherche Scientifique (CNRS), Lyon France, **3** Institut Français de Recherche sur l'Asie de l'Est (IFRAE, UMR 8043), Institut National des Langues et Civilisations Orientales (Inalco), Université de Paris, Centre National de la Recherche Scientifique (CNRS), Lille, Paris, France, **4** Institut universitaire de France (IUF), Paris, France, **5** Institute of Nomadic Archaeology and Department of Anthropology and Archaeology, National University of Mongolia, Ulaanbaatar, Mongolia, **6** Leibniz-Zentrum für Archäologie, Ludwig-Lindenschmit-Forum-1, Mainz, Germany, **7** Centre de Recherche en Archéologie, Archéosciences, Histoire (CReAAH, UMR 6566), Université Rennes 1, Centre National de la Recherche Scientifique (CNRS), Rennes, France, **8** Archéologie & Philologie d'Orient et d'Occident (OOROC, UMR 8546), Université Paris Sciences & Lettres, Centre National de la Recherche Scientifique (CNRS), Paris, France, **9** Service municipal d'Archéologie de la ville de Chartres, Chartres, France, **10** Laboratoire des Sciences du Climat et de l'Environnement (LSCE/IPSL, UMR 8212), Commissariat à l'énergie atomique et aux énergies alternatives (CEA), Centre National de la Recherche Scientifique (CNRS), Université de Versailles – Saint-Quentin-en-Yvelines (UVSQ), Gif-sur-Yvette, France, **11** Centre d'Écologie et des Sciences de la Conservation (CESCO), Muséum National d'Histoire Naturelle, Centre National de la Recherche Scientifique (CNRS), Sorbonne Université, Paris, France, **12** Centre d'Ecologie Fonctionnelle et Evolutive (CEFE), Centre National de la Recherche Scientifique (CNRS), École Pratique des Hautes Études (EPHE), Institut de Recherche pour le Développement (IRD), Université Paul Valéry Montpellier 3, Montpellier, France

* antoine.zazzo@mnhn.fr

## Abstract

Pastoral nomadism is of great cultural and economic importance in several regions of the world today. However, documenting ancient patterns of mobility in societies where pastoralism was central is challenging and requires tailored approaches and methodologies. Here we use strontium, oxygen and carbon isotopic analyses of dental enamel, together with a local strontium isoscape, to reconstruct the mobility patterns of seven domestic horses deposited in a Late Bronze Age grave from western Mongolia. Radiocarbon indicates that the animals were deposited within a short period of time, 3000 years ago. The isotope time series obtained from tooth enamel shows that four of the seven horses exhibited a common pattern characterized by a high frequency of mobility, suggesting that in this area (1) cyclical pastoral mobility dates back at least to the Late Bronze Age and (2) the animals belonged to the same herding family, implying that only a small community was involved in the funerary rite



**Data availability statement:** All relevant data are within the paper and its Supporting Information files.

**Funding:** Fieldwork was supported by the Joint French-Mongol Archaeological Mission (Ministère des Affaires Etrangères et du Développement International). This research was funded by a Ph.D. grant to NL from the French National Research Agency/Agence Nationale de la Recherche Laboratoires d'Excellence ANR-10-LABX-0003-BCDiv, by a postdoctoral fellowship to MLC from the French National Research Agency/Agence Nationale de la Recherche ANR-20-CE27-0018 and by an Inalco (Institut National des Langues et Civilisations Orientales) Early Career Research Grant 2017 to CM. There was no additional external funding received for this study.

**Competing interests:** The authors have declared that no competing interests exist.

of this structure. The data show that the other three horse individuals had a distinct mobility pattern and that one was not from the local area, pointing to flexibility in mobility patterns over time or circulation of animals between herding groups. These results illustrate the power of the isotopic approach to reconstruct animal biographies and effectively address the archaeology of pastoral nomadism and mobility.

## 1 Introduction

Nomadic pastoralism, the practice of raising livestock by moving herds cyclically to make the best use of resources and environmental conditions, is of significant importance in several regions of the world today, with tens of millions of people relying on this means of subsistence, particularly in arid and semi-arid regions [1]. Since millennia's, nomadic herding has been the cornerstone of Mongolia's culture and economy, currently supporting currently the livelihoods of about a third of the population and contributing significantly to the national economy [2]. Herding livestock, primarily sheep, goats, cattle, horses, and camels, provides nomadic families with food, transformable materials (wool, hair, hide), fuel (dung) and labor forces (riding, pack and draft animals), which they also sell and export to generate income. Mounted pastoralism was adopted in Mongolia at the beginning of the Late Bronze Age (c. 1200 BCE) and played a significant role in shaping the social, economic and cultural landscape of the region [3,4]. As a domesticated animal, the horse is thought to have had a strong influence on the adoption of pastoral nomadism by facilitating long-distance movements between pastures [5]. It also played a leading role in ritual practices. Its importance is clearly visible in the archaeological record, especially in the construction of monuments, the ritual and mortuary complexes called deer stones and khirgisuurs (DSK), which are abundant in central and northern Mongolia, but also (however in smaller numbers) in the foothills of the Altai [3,6–10]. Stone mounds are arranged around a burial mound enclosing a human body or are associated with standing stones engraved with images of deer. Under these mounds, elements of the head (skull, jaw), neck (one or more cervical vertebrae, and sometimes terminal phalanges of horses) are found. Some khirgisuurs are quite small containing fewer than ten horse head mounds, while others can reach hundreds of horse head mounds [6,11,12]. Recent analyses have facilitated significant advancements in our understanding of the gestures that precipitated the deposition of these horse heads, including but not limited to cutting, consumption, handling, and arrangement [12,13]. Although the frequency and timing of these deposits remain unknown, isotopic analysis shows that the horses did not all die at the same time of year [14], ruling out a single sacrificial event at the site. Bayesian modelling of radiocarbon dates suggests that the deposits span about fifty years in large khirgisuurs [11]. Although the exact meaning of such deposits remains difficult to determine, they could be related to the transportation of the deceased to the afterlife (like a funeral carriage) as part of the tribute paid to him by the community [8]. But these imposing deposits also raise the question of the origin and mobility of these horses, with some possibly having distant origins through gifts, trade or exchange in relation to their social significance [9,15].

While archaeological evidence provides insights into the lifestyles of Bronze Age pastoralists through their material culture, documenting other aspects of their lives, such as herd and pasture management and their mobility patterns, presents several challenges. The transient nature of seasonal campsites leaves minimal architectural remains, making it difficult to identify and excavate their sites [16]. Furthermore, the dispersed nature of nomadic populations across vast landscapes complicates systematic survey and excavation efforts [17]. These specific challenges highlight the need for tailored research approaches and methodologies to effectively address the study of pastoral nomadic mobility in the past.

Multi-isotope (Sr, O, C) analysis of archaeological remains is a well-established method for reconstructing features of human and animal life histories [18–20]. As tooth enamel is not remodeled once fully mineralized, sequential sampling along the growth axis of hypsodont (high-crowned) enamel provides a chronological record of changes in dietary habits, climatic conditions and mobility over the course of tooth development [21–23]. Carbon isotope composition ($\delta^{13}C$) reflects the $\delta^{13}C$ values of ingested plants and varies mainly according to the relative abundance of $C_3$ and $C_4$ plants in the diet [24]. Indirectly, changes in $\delta^{13}C$ along the tooth may reflect mobility (e.g., altitudinal movements) through changes in the $C_3/C_4$ composition of the plant communities encountered [25–29]. Variation in oxygen isotope values ($\delta^{18}O$) is likely to reflect variation in climatic conditions, as $\delta^{18}O$ in meteoric waters is influenced by temperature, humidity, rainfall, elevation and continentality [22]. As such, it can be used to anchor other isotopes in a seasonal framework [22]. The strontium isotope ratio ($^{87}Sr/^{86}Sr$) is widely used to assess the origin of individuals or to track their mobility [30,31]. In the environment, bioavailable $^{87}Sr/^{86}Sr$ is mainly determined by the nature and age of the underlying lithology and is spatially well defined [32]. In recent years, methods have been developed to predict the spatial distribution of $^{87}Sr/^{86}Sr$ at large scales and to generate $^{87}Sr/^{86}Sr$ isotope maps, or isoscapes [33,34], relying in particular on machine learning approaches [35,36]. Isoscapes then allow the study of mobility through geographic mappings of the $^{87}Sr/^{86}Sr$ values found in the samples [34,37].

In Mongolia, a pioneering study used C, O and Sr to discuss the geographical origin of Bronze Age horses found in peripheral stone mounds of several khirgisuurs in central Mongolia and proposed that they came from geographically distant locations [15]. However, the lack of a local isoscape available for Mongolia at the time did not allow the location and movement of the animals to be assigned geographically. Furthermore, only a few individuals from very large khirgisuurs containing hundreds of horses were analyzed, preventing any discussion on herd management.

Here, we reconstruct the mobility patterns of Bronze Age pastoralists by describing the seasonal movements of their domestic horses. A multi-isotope (C, O, Sr) approach consisting of high-resolution sampling and isotopic analysis of dental enamel is used. The archaeological remains come from the site of Burgast in western Mongolia, where a small khirgisuur excavated by our team in 2016 yielded the remains of seven horses. We established the chronological framework through radiocarbon dating and Bayesian modeling of the dates. We created a regional isoscape using a global bioavailable $^{87}Sr/^{86}Sr$ dataset, supplemented by local $^{87}Sr/^{86}Sr$ plant data from the same region published by our team [38] as well as plant and soil data from the literature collected in Asia, and performed geographic assignments of the $^{87}Sr/^{86}Sr$ enamel samples, using $\delta^{13}C$ and $\delta^{18}O$ to refine and seasonally anchor the assignments. The model predictions were tested by analyzing the tooth enamel of a local modern horse. We show for the first time that Bronze Age pastoralists in western Mongolia practiced a type of mobility similar to that observed in Altai today, characterized by a cyclical repetition of geographically restricted but frequent movements. Our results also provide evidence for variability and flexibility in mobility patterns and/or circulation of animals between herders. Taken together, they suggest that small khirgisuurs were likely places where family-level practices were carried out.

## 2. Materials and methods

### 2.1. The khirgisuur of Burgast and the Late Bronze Age (LBA) horses

The site of Burgast is located in the Bayan-Ölgii Province (aimag), Nogoonnuur district (sum), about 100 km north-east from the provincial capital Ölgii, 15 km south-west from the border with the Republic of Tuva and 20 km south of the Republic of Altai (Fig 1). This area is located on the eastern fringes of the Altai Mountains range, in its Mongolian part.

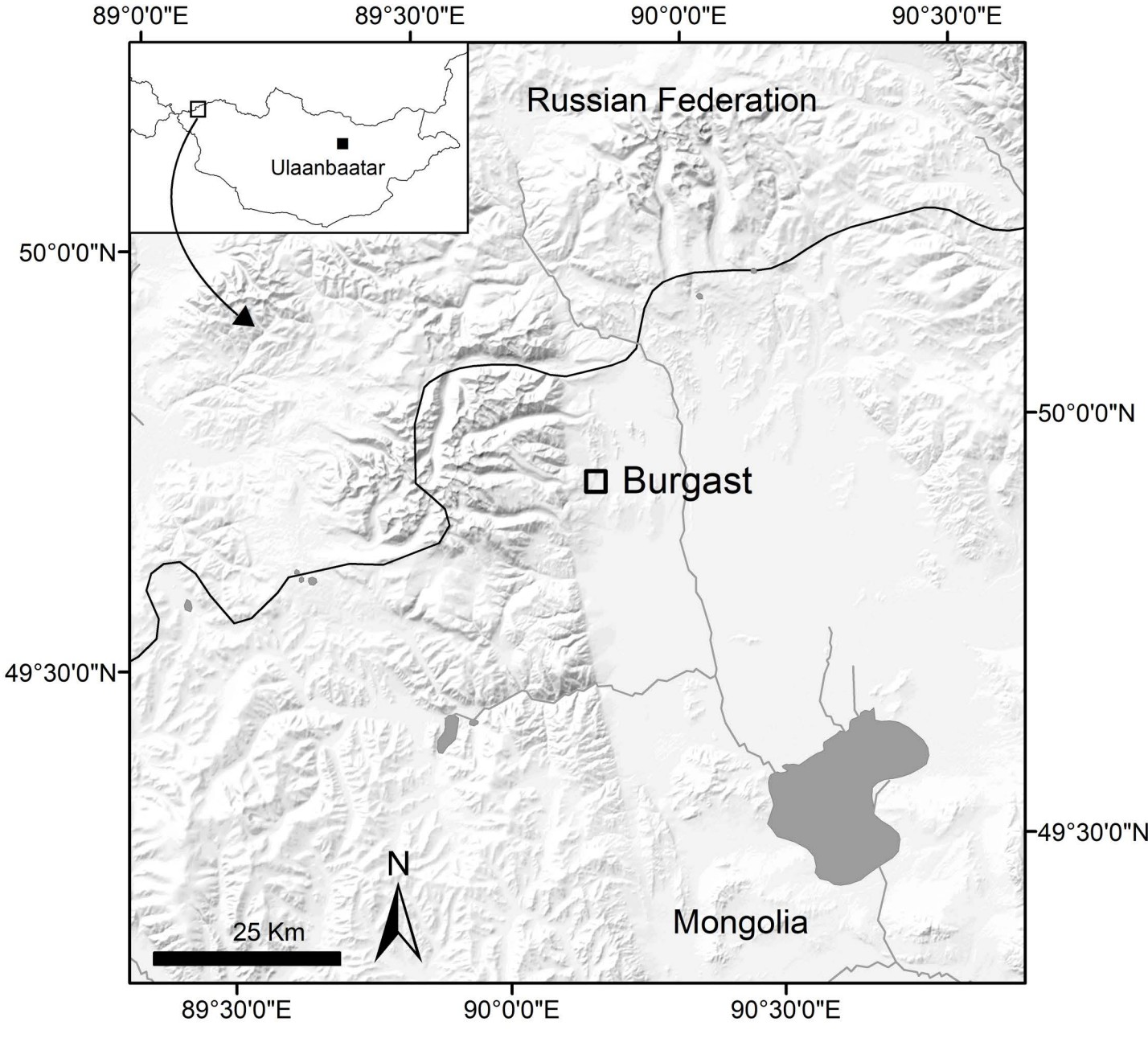

**Fig 1. Map of the study area.**

Bordered to the south by a river and to the north by a rocky massif, the site has the shape of a triangular terrace of ca. 20 ha, at an altitude of 1900 m. Archaeological remains from several periods (Late Bronze Age, Bulan Kobin culture of Xiongnu to Rouran periods and Türkic period) are scattered along the terrace, and these were excavated in 2015 and 2016 by the French-Mongolian Archaeological Mission [8]. The Late Bronze Age khirgisuur has a standard structure for this type of funerary monument found in Mongolia. It is composed of a 15.5 × 14 m central stone mound containing the remains of a human body oriented west-east (Fig 2, Fig 3). Excavation of the stones revealed the existence of a casing

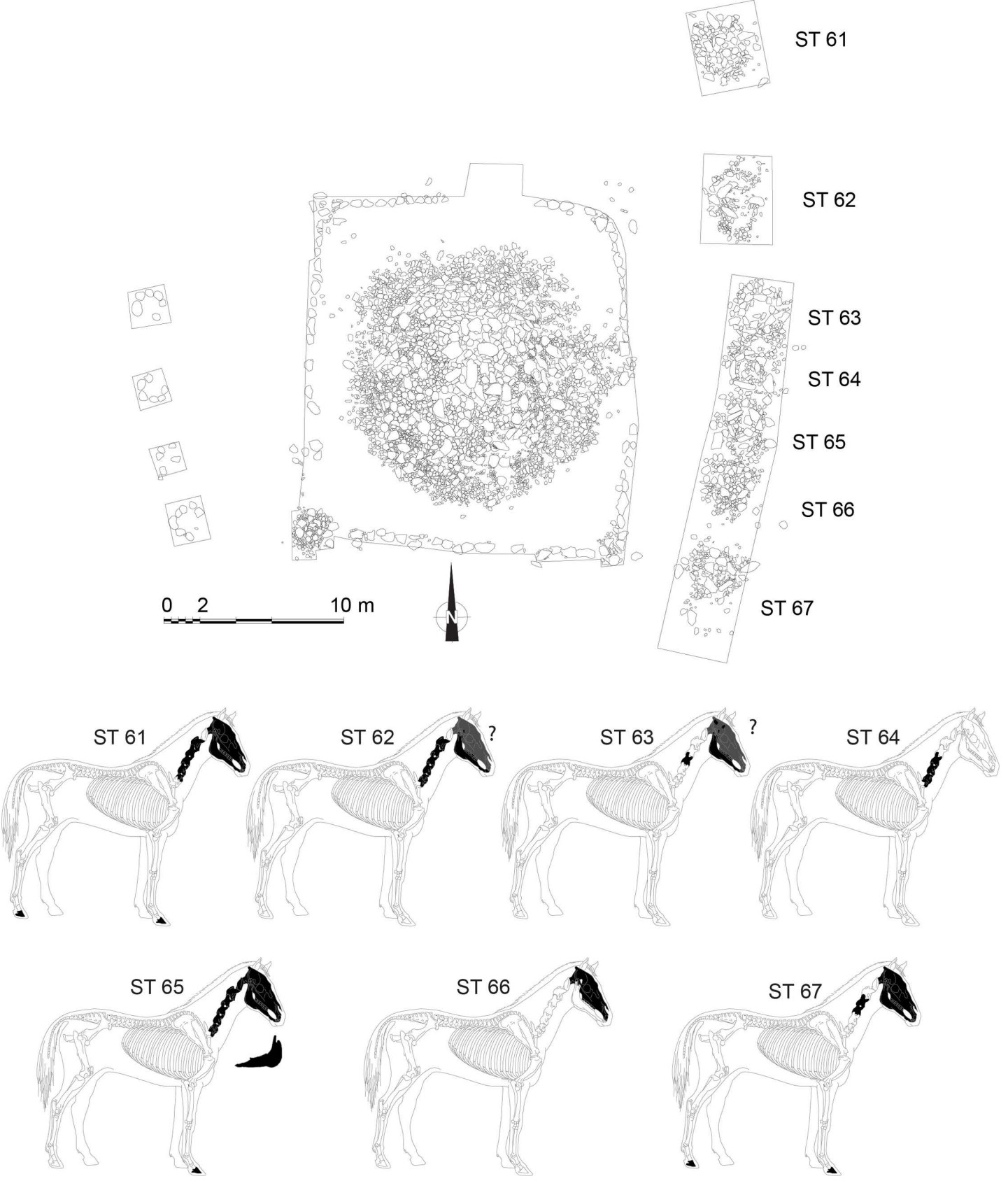

ST 61 ST 62 ST 63 ST 64 ST 65 ST 66 ST 67

**Fig 2. Layout of the khirgisuur of Burgast with structures containing horse remains, and details of the horse bone elements found in each structure.** Figure created by the author, which includes an image by Michel Coutureau (Inrap), in collaboration with Vianney Forest, © 1996 ArcheoZoo.org, used under a CC BY license.

made of large rocks. No grave goods were found in association with the deceased. A quadrangular fence (20 × 18 m) formed by an alignment of stones and having small mounds at the corners surrounds this central mound. As with most khirgisuurs, the main burial is associated with external structures in the form of mounds and stone circles. On the western side of the rectangular area, there are four stone circles aligned over a length of 14 m (ST 68–71). These stone circles did not reveal any remains, although these structures generally yield burnt bones and teeth of caprines [12,39,40]. On the eastern side, there are seven mounds (ST 61–67) yielding horse remains of 7 individuals, mainly represented by their skulls or mandibles and sometimes accompanied by connected cervical vertebrae and terminal phalanges. The horse remains are fairly well preserved, but bone displacements, breakage and missing parts seem to indicate that taphonomic disturbances (possibly by burrowing animals) have occurred, although it is not always possible to precisely define their extent. All the horse individuals were included in the study (see Table S1 in S1 File for horse details). All necessary permits were obtained for the described study, which complied with all relevant regulations. The first (M1), second (M2) and third (M3) mandibular or maxillary molars were targeted, depending on the age of the animal and the state of preservation of the tooth material, thus providing a record of the first 4–5 years of the animal's life. In modern horses, enamel mineralization occurs from 0.5 (±1) to 23 (±3) months for the M1, from 7 (±1.5) to 37 (±3) months for the M2, and from 21 (±3) to 55 (±2) months for the M3, integrating in average environmental isotopes values over 1.9, 2.5 and 2.8 years respectively [41].

One modern horse, 2017–104, was also included as a modern reference. The horse was owned by Nursultan's family (the name has been changed), Kazakh-Mongolian herders, living in the Burgast area [14,42]. The collection of 2017–104's tooth was opportunistic, its death being unrelated to the study. From the known date of death of 2017–104 [43], we estimated that the $M_2$ was formed between late 2013 and spring 2016. As part of a previous study [44], one horse of the same herder was equipped with a GPS collar (Globalstar GPS collar, Lotek Wireless Inc, http://www.lotek.com) which recorded its movements between June 2015 and August 2017. Therefore, the time period covered by the GPS collar and tooth formation of 2017–104's partially overlaps. We used the $^{87}Sr/^{86}Sr$ values observed at each GPS location, extracted from the $^{87}Sr/^{86}Sr$ isoscape to validate the $^{87}Sr/^{86}Sr$ values assessed from the modern horse enamel isotopic analysis (see below).

## 2.2. Radiocarbon dating

The bones were mechanically cleaned by abrasion using a tungsten drill tool. They were then cut into 1 gr subsamples and ground to powder, sieved to separate the 350–700 μm grains from which 150 mgs were used for collagen extraction, following Zazzo et al. [11]. The bone was demineralized in a 1M HCl solution for 20 min at room temperature. Due to the good preservation of the material, two 10 min treatments in NaOH (0.125M) were sufficient to remove any humic and fulvic acids. The samples were then solubilized in a 0.01M HCl solution for 17 h at 95°C and filtered. Finally, the gelatin samples were freeze-dried for 48 h and the collagen yield, expressed as the mass of collagen obtained per mass of processed bone sample, was calculated. An aliquot of 2.5 mg of bone collagen from each sample was wrapped in an ultra-light tin capsule and combusted in an Elemental Analyzer (EA) and then transferred to an AGE 3 automated compact graphitization system. Two oxalic acid II standards and two phthalic anhydride blanks were treated with the unknowns for every 10 samples. The reduction was carried out in seven quartz reactors, each containing 5 mg of iron catalyst. Prior to burning each sample, an aliquot of that sample was burnt to minimize cross-contamination. Following combustion, %C and N, as well as C/N ratios of each individual sample were calculated. Only samples which met the three following quality control criteria were then pressed into targets and dated. The state of collagen preservation was assessed using the criteria

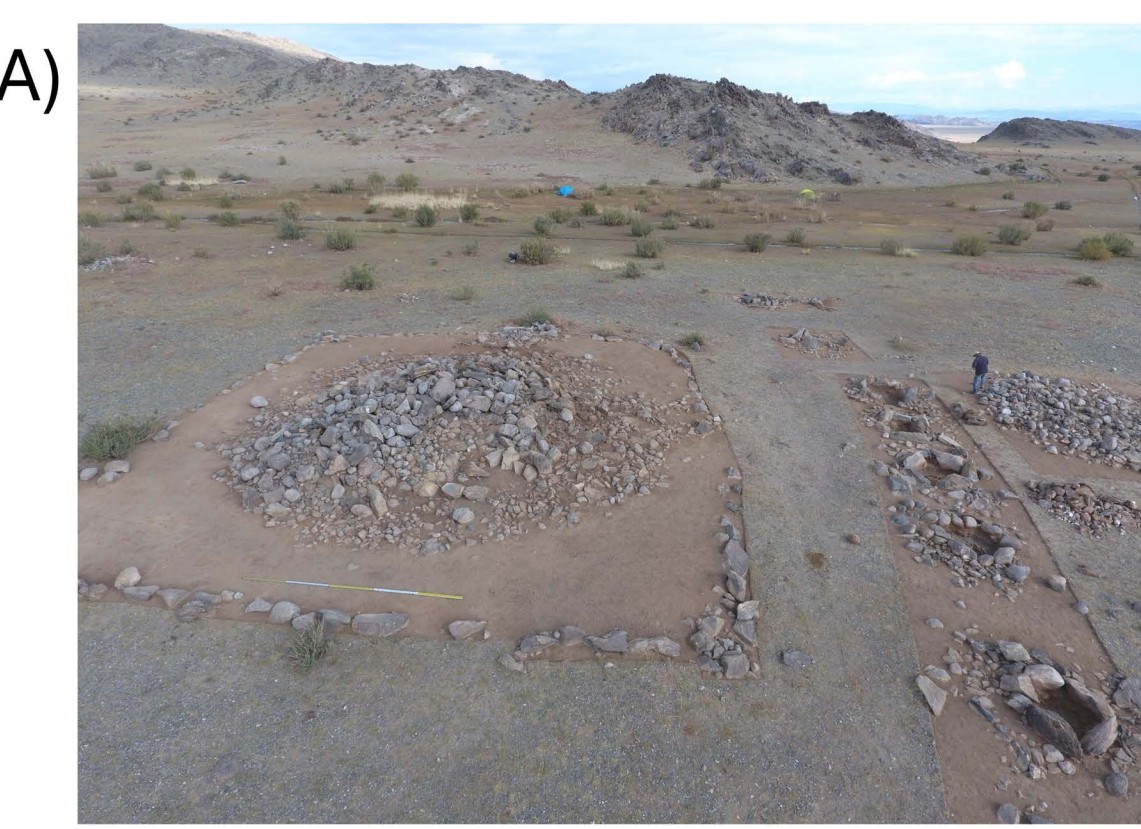

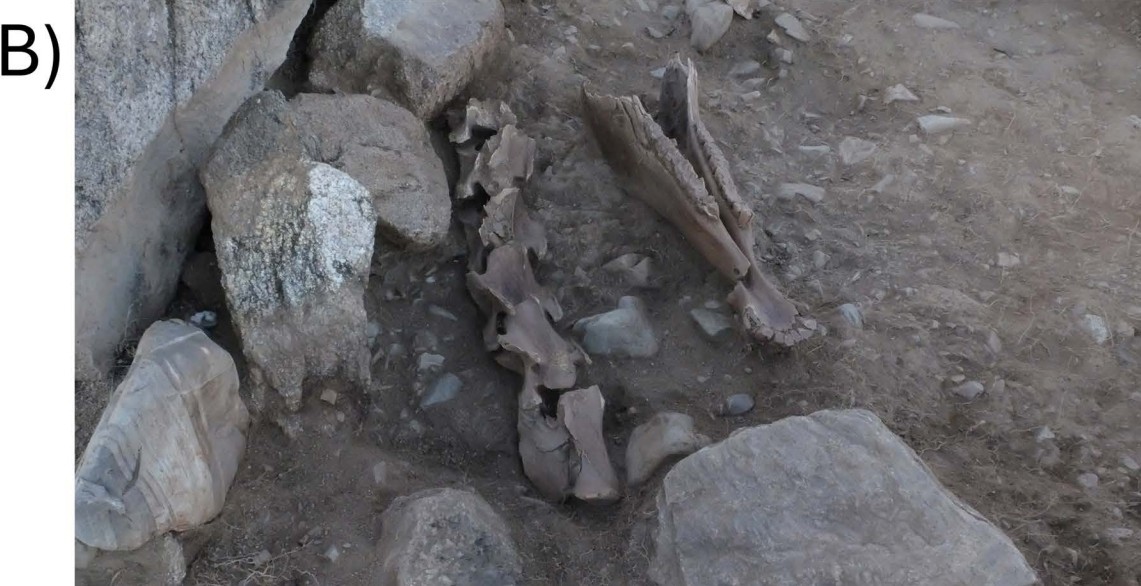

**Fig 3. Pictures of the khirgisuur.** A) Aerial **view of the khirgisuur with the stone circle on the left and structures containing horse remains on the right**. B) Details **of horse bones and teeth from ST 62.**

defined by Zazzo et al. [11] on Mongolian samples and only samples with C/N ratios between 3.1 to 3.3 or with at least 5% extraction efficiency were considered suitable for dating. The graphite targets were analyzed using the compact AMS ECHoMICADAS in the *Laboratoire des Sciences du Climat et de l'Environnement* (LSCE, UMR 8212, CNRS/CEA/UVSQ) in Saclay (France). Data reduction was performed using BATS software (version 4.07). Radiocarbon dates were calibrated using the latest Northern Hemisphere curve (NHcal20) [45] and the latest version of Oxcal software (v4.4) [46,47]. A Bayesian modeling framework consisting of a single phase containing the eight radiocarbon dates was built to determine the beginning, end and duration of the construction and use of the monument [48].

## 2.3. Carbon and oxygen isotopic analysis

Carbon and oxygen isotopes were used to investigate the life history of individuals. Changes in enamel $\delta^{18}O$ values reflect the variation in $\delta^{18}O$ of meteoric waters during tooth formation [22]. Thus, $\delta^{18}O$ can be used to set the seasonality of the teeth, the warm summer season being indicated by high $\delta^{18}O$ values, while low $\delta^{18}O$ values correspond to the cold winter season [22]. Along an elevation profile, changes in the $\delta^{13}C$ values are influenced by the changes in the relative abundance of $C_3$ and $C_4$ plants in plant communities [43,49,50]. Consequently, $\delta^{13}C$ values of enamel can be employed as a proxy for the elevation at which horses fed during the formation of their teeth [43]. An equation that links the $\delta^{13}C$ in diet to elevation has been specifically calibrated for use with horses in the Altai region, where the $\delta^{13}C$ of plants decreases as elevation increases [43].

Carbon and oxygen isotopic ratios were measured along the teeth of the modern and archaeological horses. Horse tooth surfaces were cleaned by mechanical abrasion with tungsten drill bits to remove the cement. Sequential sampling of the teeth was performed on the mesial side of the molars. The mesial side was chosen for practical convenience (this side has a flat surface with a narrow cementum layer) and also because it should mineralize faster [51]. The sampling spans the width of the mesial side. Enamel was drilled with a diamond bit, perpendicular to the tooth growth axis from the top of the crown to the enamel root junction (ERJ). Samples were taken every 1.5–2 mm through most of the entire thickness of the enamel layer and care was taken to avoid collecting the underlying dentine. Sample position was recorded in terms of their distance (in mm) from ERJ. Powdered enamel thus recovered (~ 6–7 mg) samples were chemically treated following the procedure described in Balasse et al. [52]. First, organic matter was removed with a solution of 2–3% NaOCl (24 h, 0.1 ml solution/ mg of sample) and rinsed several times in distilled water. Then, each enamel sample reacted with 0.1 M acetic acid (4 h, 0.1 ml solution/mg of sample) to remove exogenous carbonates and rinsed in distilled water and dried in an oven at 80°C for 12h. Purified enamel samples were weighted (600 μg approximately) and analyzed on a Kiel IV device connected to a Delta V Advantage IRMS at the *Service de Spectrométrie de Masse isotopique du Muséum national d'Histoire naturelle* (SSMIM) in Paris. Samples were placed into individual vessels to react under vacuum with phosphoric acid [$H_3PO_4$] at 70°C and purified $CO_2$ in an automated cryogenic distillation system. Analytical precision was 0.04‰ for $\delta^{18}O$ and 0.02‰ for $\delta^{13}C$ based on the repeated (n = 30) analysis of our internal laboratory carbonate standard (LM marble) normalized to NBS 19. Isotope data are presented in δ notation [$\delta = (R_{sample}/R_{standard}) - 1$], with $R$ the isotope ratio [$^{18}O/^{16}O$ and $^{13}C/^{12}C$] of the sample, reported relative to the international standard V-PDB (Vienna PeeDee Belemnite) and expressed in per mil (‰).

## 2.4. Enamel $^{87}Sr/^{86}Sr$ analysis

To investigate the mobility of modern and ancient horses, we sampled the $^{87}Sr/^{86}Sr$ in teeth enamel along the enamel-dentine junction from the occlusal surface (OS) to the ERJ of the molars using Laser Ablation Multi-Collector Inductively Coupled Plasma Mass Spectrometry (LA-MC-ICP-MS). Following $\delta^{18}O$ and $\delta^{13}C$ analysis, the teeth were washed in distilled water and then were sliced longitudinally with a diamond disc, mounted in polyester resin (GBS - BROT LAB) to expose the whole dental structure and polished (Abrading with silicon carbide powder 13 and 5 μ – F500 and F1000 Escil®). We targeted the inner enamel layer, because this zone is highly mineralized early in the mineralization process

which limits the attenuation of the isotopic signal [21,53–55]. When needed, laser ablation profiles were performed mainly on the lingual side to avoid the pits caused by $\delta^{18}O$ and $\delta^{13}C$ analysis. $^{87}Sr/^{86}Sr$ was measured *in situ* by LA-MC-ICP-MS. Sr data were acquired with a Thermo-Fisher™ Neptune Plus™ (Thermo Fisher Scientific), coupled to a 193 nm laser ablation system – (Elemental Scientific Lasers - NWR 193 system) at the *Ecole Normale Supérieure* in Lyon (ENSL, [56]). The inner enamel layer was ablated using 100 μm laser spot size, a laser scan speed of 60 μm.s$^{-1}$, a fluence of 11 J/cm² and 10-Hz frequency (Table S2 in S1 File summarizes the operating parameter used for the LA-MC-ICP-MS measurements). Between each tooth enamel profile, calibration was achieved by analyzing NIST SRM 1400 (bone ash). Sample aerosol was carried to the ICP-MS using a mixture of He and $N_2$. Masses 88, 87, 86, 85, 84 and 83 were measured on Faraday cups. The isobaric interference of $^{87}Rb$ was corrected using the $^{85}Rb$. Krypton interferences at masses 84 and 86 were corrected using $^{83}Kr$. The Rb- and Kr-corrected $^{87}Sr/^{86}Sr$ ratio was further corrected by a standard-sample-standard bracketing method using a pellet of sintered NIST SRM-1400 certified reference material as the standard (bone ash, [57]). Repeated measurements of the sintered reference material NIST SRM-1400 gave an average $^{87}Sr/^{86}Sr$ value of 0.71401 ± 0.00165 (± 2SD, n = 27) in good agreement with the published values for this material using Thermal Ionization Mass Spectrometry (0.71308 ± 0.00002, ±2SD, n = 25, [58]). The bracketing correction was applied for each $^{87}Sr/^{86}Sr$ value depending on its position in the transect relative to the preceding and following bracketing standards. The average $^{88}Sr$ voltage (3.5 ± 0.8 V, ±2SD, n = 25) measured on the sintered reference material SRM1400 that contains 249 mg/g of Sr, yields an overall transmission of 14 mV/ppm, considering the above-mentioned laser settings. This performance is in good agreement with other studies with comparable settings [59]. Finally, a centered moving average on 35 individual measurements (~2 mm length) was used to smooth variability of the final profiles.

## 2.5. Statistical analysis

The average $\delta^{18}O$, $\delta^{13}C$ and $^{87}Sr/^{86}Sr$ values between individuals were compared using the One-way ANOVA statistical test coupled with Tukey's post-hoc test using R software v4.3 [60] with a significance threshold at p-value < 0.05. The $\delta^{13}C$ measured in the teeth were converted to elevation following Lazzerini et al. [43]. First, $\delta^{13}C$ values from enamel were corrected by -2‰ to account for the Suess effect [61]. We then estimated the $\delta^{13}C$ of diet ($\delta^{13}C_{diet}$) from the $\delta^{13}C$ observed in tooth enamel using a fractionation between diet and enamel of -13.7‰ assessed for horses [62]. Finally, we applied the equation from Lazzerini et al. [43] to calculate elevation from $\delta^{13}C_{diet}$:

$$\delta^{13}C_{diet} = -1.45e^{-03}\left(\pm 3.07e^{-04}\right) \times \text{Elevation} - 24.14\left(\pm 0.71\right) \tag{1}$$

The average elevation between individuals was compared using a one-way Anova coupled with a Tukey's post-hoc test.

## 2.6. Strontium isoscape

We generated a bioavailable $^{87}Sr/^{86}Sr$ isoscape for Mongolia and more specifically for the Altai region, updating an already existing global bioavailable $^{87}Sr/^{86}Sr$ isoscape from Bataille et al. [33] with local $^{87}Sr/^{86}Sr$ data from plants. Designed to predict the spatial distribution of $^{87}Sr/^{86}Sr$ across the world, the original model [33], based on random forest (RF), a machine-learning algorithm, and on a database of more than 4000 sampling site, lacks, however, precision and accuracy in Asia and Mongolia due to insufficient sampling. To improve predictions, we incorporated 156 local plant data from the Altai region [33,36,38] along with recent plant and soil data from Asia listed in Table S8 in S2 File.

To predict the spatial distribution of the bioavailable $^{87}Sr/^{86}Sr$, the RF integrates lithological, environmental and climate variables observed at each sampling site that likely influence the $^{87}Sr/^{86}Sr$ value in the environment [33]. Following Le Corre et al. [36], we used minimal/maximal age and predicted $^{87}Sr/^{86}Sr$ value of the bedrock [63], terrane age, topography, soil properties, climate variables and salt and dust deposits. The list and details of all the predictors are provided in Table

S4 in S1 File. Using R v4.3 [60], we extracted these variables at each $^{87}$Sr/$^{86}$Sr sampling site. RF was then applied to the $^{87}$Sr/$^{86}$Sr sampling site dataset and the associated covariates, using data from soil, plant and animal with limited movement range samples to ensure local $^{87}$Sr/$^{86}$Sr values [33,36].

RF grows multiple regression trees by bagging [64]: for each tree, the dataset is divided by bootstrap into a training dataset and a validation dataset or "out-of-bag". A regression tree is grown on the training dataset, and the validation dataset is used for internal cross-validation. The outcome of each tree is then aggregated to obtain the predictions. RF does not make assumptions on data distribution and homoscedasticity, allowing handling complex relationships in the data [64]. We conducted our analysis in R v4.3 [60], following Le Corre et al. [36], we filtered highly correlated variables (R > 0.9) and used a variable selection algorithm (*VSURF* R package [65]) to select the most relevant predictors. The final model was trained with 3000 trees and evaluated using ten-fold cross validation [64]. The model was then applied to covariate rasters to predict $^{87}$Sr/$^{86}$Sr across Mongolia and a quantile random forest regression was used to generate a 68,27% prediction interval and estimate the spatial uncertainty (standard deviation) associated with the prediction [66]. More details about the method are provided in supplementary material SM4.

### 2.7. Geographic assignment

In order to infer the mobility of horses from the $^{87}$Sr/$^{86}$Sr intra-tooth profiles of both the Bronze Age and modern horses, a Bayesian geographic assignment [67] was performed on samples taken along the $^{87}$Sr/$^{86}$Sr intra-tooth profiles [61]. The Bayesian approach is particularly well-suited for geographic assignment on continuous-probability surfaces as it inherently accounts for the probabilistic nature of the model and the associated uncertainties [67].In R, we used the *assignR* package [68] to determine the geographic origin in western Mongolia of the horses from the Burgast archaeological site. For each $^{87}$Sr/$^{86}$Sr intra-tooth profile (n = 15), we computed a rolling average and a rolling standard deviation using a window of 35 samples to smooth the signal. We then identified features on the profiles (e.g., peaks, troughs, plateaus) representative of the variation in $^{87}$Sr/$^{86}$Sr for each tooth and carried out assignment analyzes on them. We treated the $^{87}$Sr/$^{86}$Sr samples from teeth as samples of unknown origin. For each sample, we generated a posterior probability surface and extracted the 10% of the area of interest with the highest probability of origin. As the probability of origin is relative to the area where the assignment is done, the assignment was initially conducted on a 200 by 200-kilometer area centered on the Burgast archaeological site to identify local and non-local individuals. We then performed a second assignment at the local scale (100*100 km) and at a larger scale (400*400 km), both areas centered on Burgast, to explore local and regional geographic assignments. Combining $^{87}$Sr/$^{86}$Sr assignment with other isotopes allows discriminating between potential areas of origin [66,69]. Here we used the δ$^{13}$C as a proxy of elevation [43] to help interpret the assignment of the samples on the $^{87}$Sr/$^{86}$Sr isoscapes. However, given the low precision of the relationship between δ$^{13}$C and elevation [43], we restricted the use of δ$^{13}$C to qualitative interpretation rather than combining assignment and elevation maps.

## 3. Results

### 3.1. Radiocarbon dating and Bayesian modeling

The ages measured in human and horse collagen samples cluster tightly between 2815 ± 30 and 2885 ± 25 BP (Table S3 in S1 File). Bayesian modeling of the dates (Fig 4) suggests that the deposition of the human and animal remains started sometime during the 11$^{th}$ c. BCE (1104–1006 cal. BCE, median 1042, 95.4%) and ended between the second half of the 11$^{th}$ c. BCE and the first half of the 10$^{th}$ c. BCE (1046–939 cal. BCE, median 1005, 95.4%). The overall span of use of the complex is between 0–104 y (95.4%) with a median value of 25 y.

### 3.2. Strontium isoscape

The bioavailable $^{87}$Sr/$^{86}$Sr isoscape, along with the associated spatial uncertainty, is presented in Fig 5. The RF model demonstrates a root mean square error (RMSE) of 0.0035, which quantifies the mean deviation between predicted

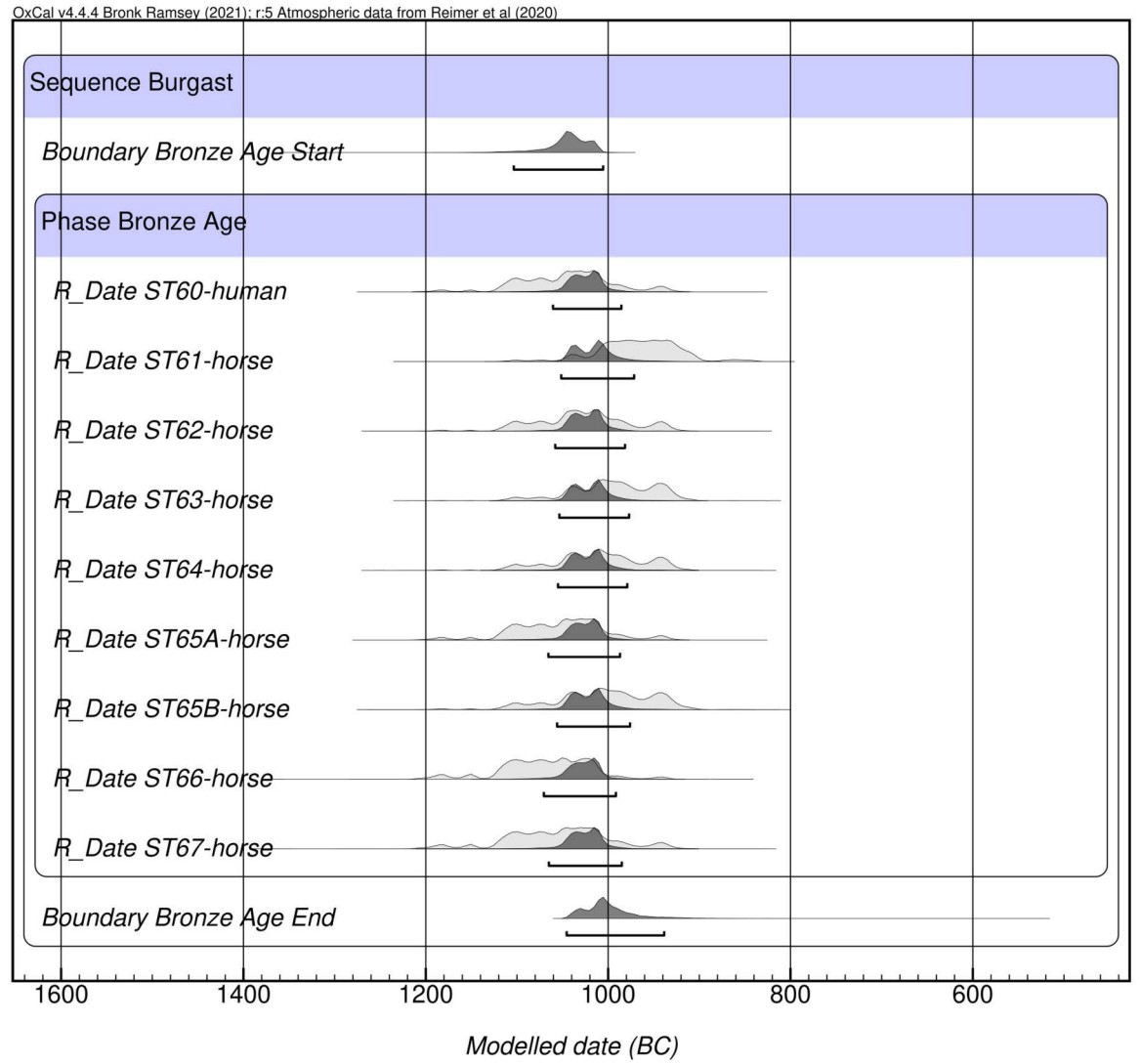

OxCal v4.4.4 Bronk Ramsey (2021); r:5 Atmospheric data from Reimer et al (2020)

**Fig 4. Bayesian phase model of the Burgast khirgisuur.**

and actual values, and an explanatory power of 61.6% of the variance. Running the model of Altai data only lead to a slightly improved RMSE (0.0022) but with a lower explanatory power of 37.1% of the variance. The final RF model was constructed using eleven predictors (Fig S1 in S1 File), including lithological variables (minimum and maximum age of the bedrock, predicted $^{87}$Sr/$^{86}$Sr value of the bedrock), climate variables (mean annual precipitation and temperature, potential evapo-transpiration), and sea salt and dust deposition. In the Altai region, the $^{87}$Sr/$^{86}$Sr exhibits a range of values between 0.7100 and 0.7195 (Fig 5C), with intermediate values (0.712–0.714) and high values also observed. Values between 0.714 and 0.716 are observed in the northern and western regions in the vicinity of Burgast, while lower $^{87}$Sr/$^{86}$Sr values are present in the southern regions, along the Altai Mountains range, with a $^{87}$Sr/$^{86}$Sr below 0.712. The $^{87}$Sr/$^{86}$Sr values from the plants sampled in Altai demonstrate a good agreement with the $^{87}$Sr/$^{86}$Sr values from the iso-scape at the sampling sites (RMSE = 0.0015, R² = 0.67), indicating that the $^{87}$Sr/$^{86}$Sr value predicted by the RF in this region is reliable.

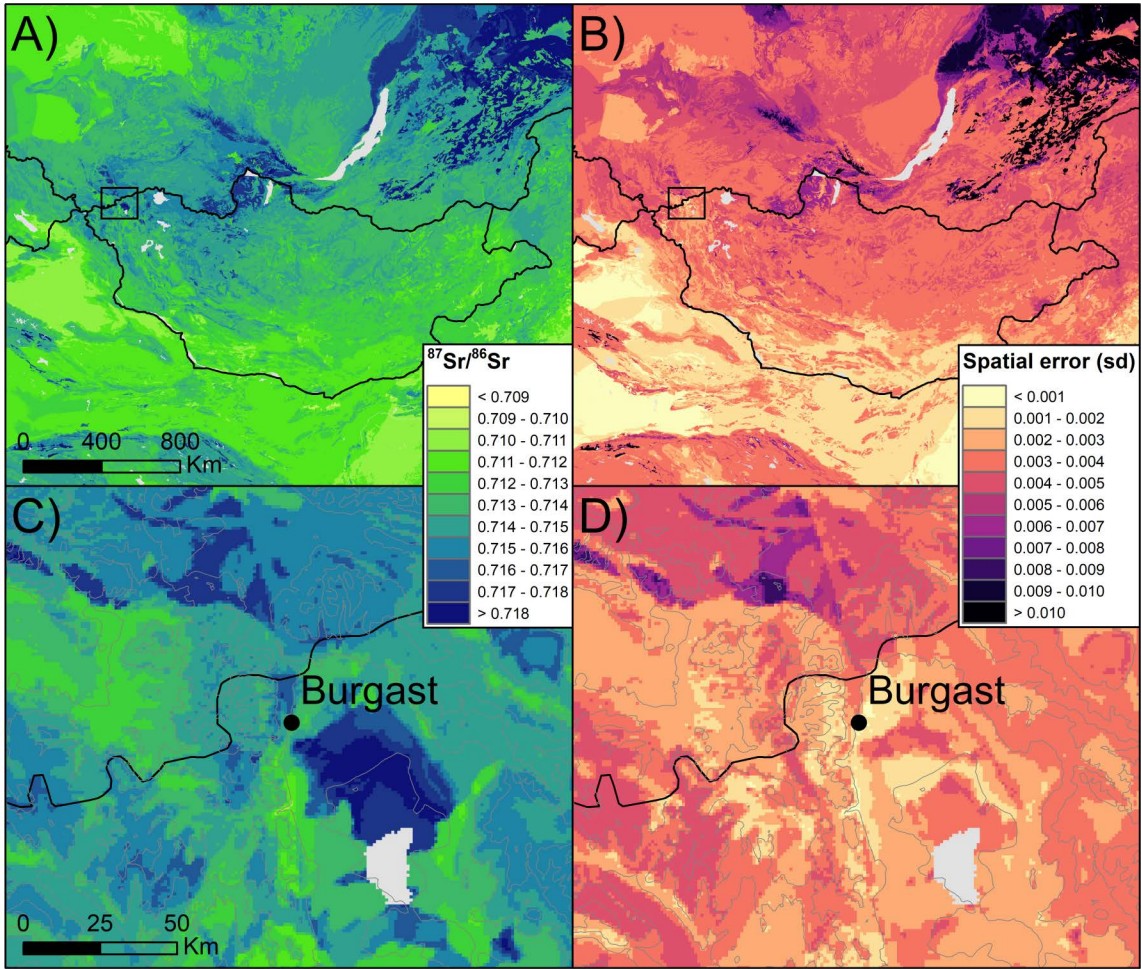

**Fig 5. Bioavailable ⁸⁷Sr/⁸⁶Sr isoscape of Mongolia.** A) predicted bioavailable ⁸⁷Sr/⁸⁶Sr for Mongolia with B) the associated spatial uncertainty, and C) and D), their respective zoom on the Altai region in western Mongolia. The zoomed areas are indicated by the square on the maps of Mongolia. The gray lines on the zoomed maps represent elevation.

### 3.3. Enamel δ¹⁸O and seasonality

Table S5 in S1 File presents the summary statistics for δ¹⁸O and δ¹³C values of horse tooth enamel. The supplementary material SM5 provides detailed inter-individual comparisons based on the post-hoc Tukey test (Fig S2 in S1 File). The mean δ¹⁸O value for the individuals is -12.8 ‰ (± 2.13 SD), with a range from -18.0 ‰ to -5.7 ‰ (Table S5 in S1 File). The mean amplitude in δ¹⁸O is 4.9 ‰ (± 1.49). Sinusoidal patterns are evident in the intra-tooth profiles of horses ST63, ST65a, ST65b, ST66, 2017–104, and, to a lesser extent, ST61 (Fig 6). The succession of peaks and troughs indicates the recording of isotope data over a period of one year (e.g., ST65a M2, Fig 6D) up to 2.5 years (e.g., ST63 M3, Fig 6C). In the case of older horses (ST62 and ST67), the presence of substantial wear on the teeth precludes the observation of cycles, yet discernible variations remain.

### 3.4 Enamel δ¹³C and altitude

The estimated elevation from tooth enamel δ¹³C values is, on average, 1562 m (± 436 m), with a mean amplitude of 939 m (±347 m; Table S6 in S1 File). Significant variation in elevation was observed between individuals (Anova: $F_{7,430} = 49.3$,

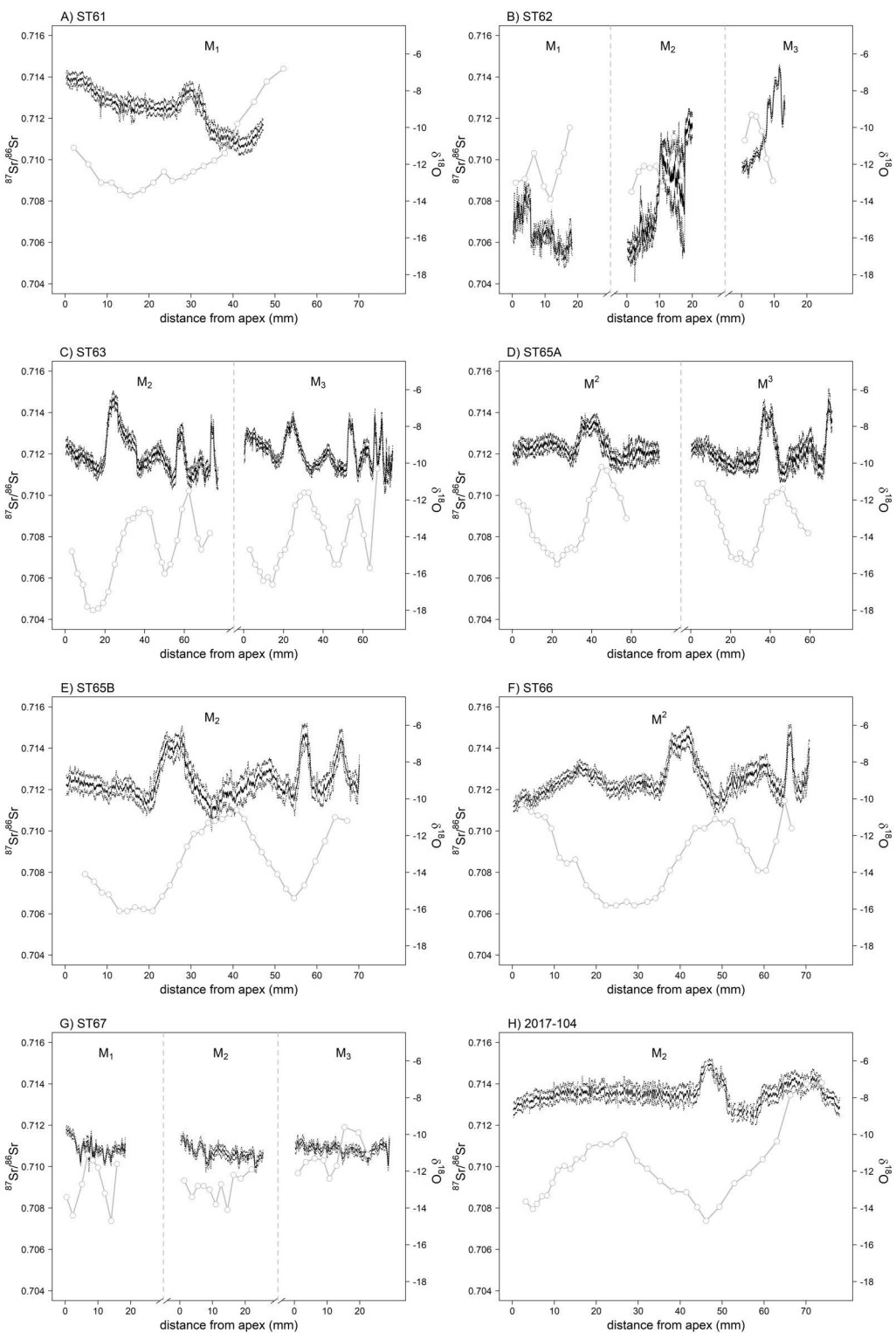

**Fig 6.** $^{87}$Sr/$^{86}$Sr and δ$^{18}$O intra-tooth profiles of the seven Bronze Age horses from Burgast (A-G) and the modern horse 2017–104 (H). $^{87}$Sr/$^{86}$Sr values were obtained by LA-ICP-MS and are presented as the rolling mean (solid black lines) with rolling standard deviation (dashed black lines) of the measurements. δ$^{18}$O data (gray lines) were obtained by IRMS analysis.

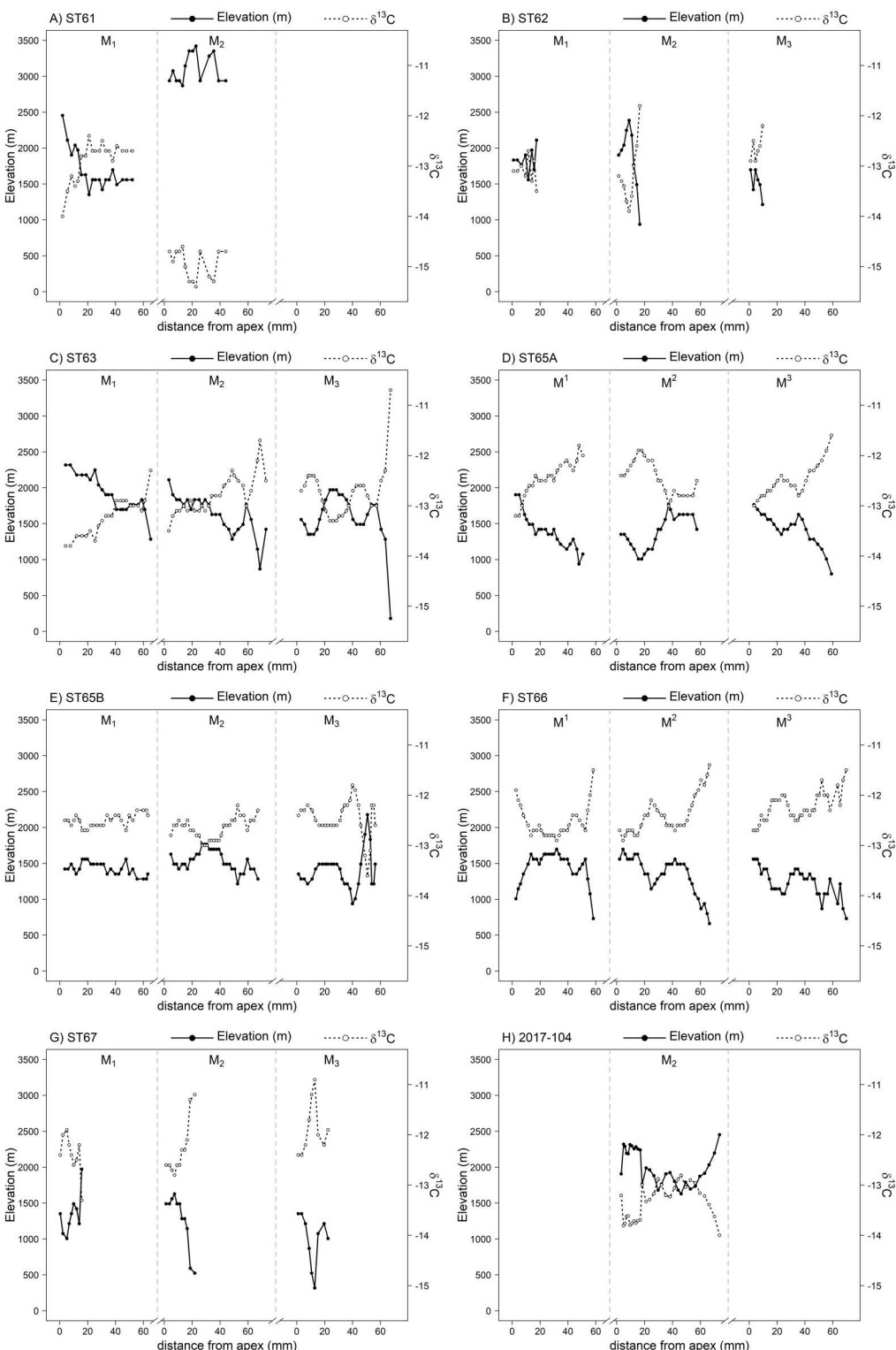

**Fig 7. Estimated elevation used by the seven Bronze Age horses from Burgast (A-G) and the modern horse 2017-104 (H) and their corresponding δ13C intra-tooth profiles.** The elevation is indicated by solid lines and the δ13C intra-tooth profiles from which elevation was estimated by dashed lines.

p-value < 0.001). Predicted elevation from $\delta^{13}C$ profile of horse ST61 are higher elevation usage compared to the other horses (post-hoc Tukey test, Fig S2C in S1 File), with predicted values for the $M_2$, exceeding 2800 m (Fig 7). Elevation predicted for ST65a, ST65b, ST66, and ST67 horses is notably similar (Table S6 in S1 File), yet significantly lower than that observed in the other horses (post-hoc Tukey test, Fig S2C in S1 File). With the exception of the modern horse, the conversion of the enamel $\delta^{13}C$, particularly from samples situated in the vicinity of the enamel-root junction (ERJ), has the potential to result in a predicted elevation below 1500 m, which is implausible for the Altai region. It is important to exercise caution when considering these values, as the uncertainty associated with the relationship between $\delta^{13}C$ and elevation may result in the generation of implausible estimates [43]. Therefore, we used intra-tooth variation in $\delta^{13}C$ values as an indication of altitudinal movements but did not use it to infer specific elevation.

### 3.5. Enamel $^{87}Sr/^{86}Sr$ and mobility

Table S7 in S1 File presents a descriptive statistical analysis of the $^{87}Sr/^{86}Sr$ intra-tooth profiles, based on rolling averaged values. For purposes of comparison between individuals, the relevant data are provided in supplementary material (SM5, Fig S2D, S3 in S1 File). Fig 8 illustrates two examples of geographic assignments of samples taken along the tooth from Bronze Age horses (ST62 and ST63), which are representative of the diverse $^{87}Sr/^{86}Sr$ values and variations observed among the intra-tooth profiles. For a given sample, the top 10% most probable origin areas are displayed on the map with elevation overlaid within these areas for interpretation, as altitudinal movements are a key aspect of pastoralism in this region of Mongolia [43]. Further detailed assignments are provided in the supplementary material for the modern local horse and three Bronze Age horses (Fig S4–S7 in S1 File). Additionally, assignments of all samples taken along the teeth for each individual are provided as short video clips (supplementary material SM9). Three distinct types of mobility can be identified from the intra-tooth isotope profiles (Fig 6) and their corresponding geographic assignments (Fig 8, S4–S7 in S1 File).

**3.5.1 Local horses displaying seasonal movements.** The $^{87}Sr/^{86}Sr$ values in the $M_2$ of ST63, ST65a, ST65b, and ST66 exhibit a series of peaks (approximately 0.713–0.714) separated by longer intervals of lower $^{87}Sr/^{86}Sr$ values (approximately 0.711–0.712). Such $^{87}Sr/^{86}Sr$ values can be observed on the isoscape within 50 km of Burgast (Fig 5). Elevated $^{87}Sr/^{86}Sr$ values observed at peaks (sample A, Fig 8A) are assigned to the immediate vicinity of the Burgast archaeological site and to an area 50 km south of Burgast along the Altai Mountains range (Fig 8B). Low $^{87}Sr/^{86}Sr$ values observed at troughs (sample B, Fig 8A) are assigned to areas south of Burgast (Fig 8B), which are estimated to be 5–15 km away from the archaeological site. At the regional scale, these areas consistently emerge as potential areas of origin as well (Fig 8B), which strongly suggests that the individual in question was local. The local map (100*100 km, Fig S8C in S1 File) further illustrates the spatial segregation between high and low $^{87}Sr/^{86}Sr$ potential origins, as well as the regularity of movements between Burgast and the area 15 km away in the south. The three peaks observed on the $^{87}Sr/^{86}Sr$ intra-tooth profile of ST63, located at a distance of approximately 20 mm, 60 mm and 70 mm from the apex, appear to mirror peaks in $\delta^{18}O$, occurring slightly before along the teeth (Fig 6C). While the values in $^{87}Sr/^{86}Sr$ decrease, the values of $\delta^{18}O$ begin to rise (Fig 6C). This similarity between the two signals suggests that changes in $^{87}Sr/^{86}Sr$ values are driven by seasonal movements. Using the $\delta^{18}O$ sinusoidal variation as a temporal marker we can observed four primary relocations over the course of a year: a first relocation to the high $^{87}Sr/^{86}Sr$ area followed by a return to the low $^{87}Sr/^{86}Sr$ area, then a relocation to an area with intermediate $^{87}Sr/^{86}Sr$ before a last return to the low $^{87}Sr/^{86}Sr$ area (Supplementary material SM9). Similar patterns are observed in the case of horses ST61, ST65a, ST65b, and ST66 (Fig 6, Supplementary Material SM9), suggesting these individuals are also local and exhibit seasonal movements.

The local modern horse (2017–104) exhibits a comparable $^{87}Sr/^{86}Sr$ profile, albeit with slightly elevated $^{87}Sr/^{86}Sr$ values and a more pronounced offset between the peaks in $^{87}Sr/^{86}Sr$ and in $\delta^{18}O$ (Fig 6H). The assignments at the local and intermediate scales are analogous to those of the local Bronze Age horses and suggest probable areas of origin for high and intermediate values of $^{87}Sr/^{86}Sr$ within a 25 km radius around Burgast (Fig S7 in S1 File). These areas are located in

## A) $^{87}Sr/^{86}Sr$ profiles

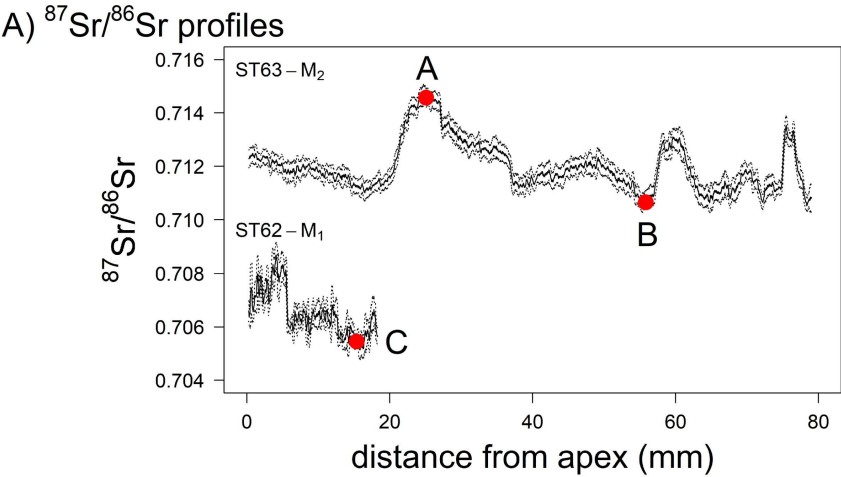

## B) Geographic assignments

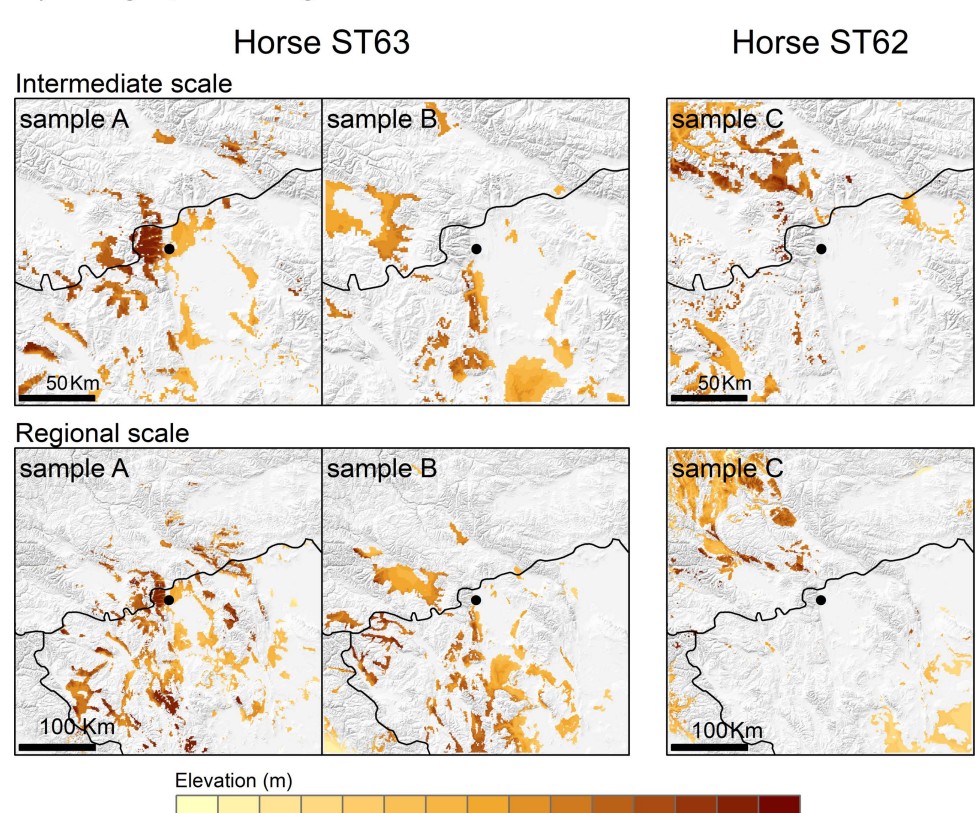

**Fig 8. Example of geographic assignment of the Bronze Age horses from ST62 and ST63.** A) the $^{87}Sr/^{86}Sr$ intra-tooth profiles of the two horses with the 3 samples selected for assignment (red dot). B) the assignment maps of the 3 samples at the intermediate scale (200*200 km) and the regional scale (400*400 km) centered on the archaeological site of Burgast (black dot). Assignments correspond to the 10% assignment map surface with the highest probability of origin. Elevation within each assigned area is displayed using a color scale.

the lowland region at the foot of the Altai Mountains range, as well as in the mountains, and are consistent with the known location of the campsites used by the herder's family (Fig S9 in S1 File). The extraction of $^{87}Sr/^{86}Sr$ values from the isoscape at each location of the modern horse equipped with a GPS collar yielded an $^{87}Sr/^{86}Sr$ profile analogous to that of the 2017–104 $^{87}Sr/^{86}Sr$ intra-tooth profile, albeit with slightly elevated values (Fig S9 in S1 File).

**3.5.2. Local horse with no visible seasonal movement.** Horse ST67 exhibits a distinct profile, characterized by a uniform $^{87}Sr/^{86}Sr$ signal across its three molars (Fig 6G), with an average value of 0.7108 (± 0.0003) and $^{87}Sr/^{86}Sr$ values ranging from 0.7100 to 0.7118. Assignments of samples from the three molars lead to nearly identical assignment maps (Fig S5B in S1 File) with more probable areas of origin observed south of Burgast, suggesting a local individual.

**3.5.3. Non-local horse.** Finally, horse ST62 exhibit the lowest values of all individuals, while also displaying the greatest variability in $^{87}Sr/^{86}Sr$ values (Fig S3 in S1 File). They range from 0.7051 to 0.7145. They show a decrease from 0.7087 to 0.7051 along the M1, followed by a sharp increase along the M2, reaching values between 0.710 and 0.712, comparable to those observed for the other horses. The $^{87}Sr/^{86}Sr$ values continue to increase along the M3, reaching a maximum of 0.7145. The low $^{87}Sr/^{86}Sr$ values observed on the M1 (sample C, Fig 8A) and at the upper part of the M2 are not observed in the Altai region (Fig 5C) and indicate an individual that originated from a location more than 50 km from Burgast. A broader scale (Fig 8B) reveals areas within a 100 km radius north and south-west of Burgast. As an alternative, areas located 200 km to the west of the Altai Mountains range or in the south suggest a more distant origin. The elevated $^{87}Sr/^{86}Sr$ values observed in the lower portion of the M2 and M3 are more consistent with the local $^{87}Sr/^{86}Sr$ signature of the Altai region. The assignments of the samples from the M2 and M3 (samples B and C on Fig S6A in S1 File) at intermediate and regional scales (Fig S6B, S6C in S1 File) highlight respectively areas south of Burgast and in the direct vicinity of the site, such as for the local horses.

## 4. Discussion

In both contemporary and historical pastoral systems observed across the Eurasian steppes, a variety of nomadic practices can be observed across different regions. Distances and seasonal mobility exhibit considerable variation, spanning a range of several kilometers to over one thousand [10,70–72]. These practices manifest in different forms, influenced by a complex set of environmental, climatic, cultural, and socio-political factors. In contemporary Mongolia, anthropologists and geographers have identified three primary categories of nomadic systems: desert, steppe and mountain nomadism [73–76]. Mountain nomadism, in Western Mongolia, is distinguished by the presence of high-altitude summer camps which are inaccessible the remainder of the year and low-altitude camps during the cold season, situated at the foot of mountains and protected from winds. The data presented here suggest an alternation between high-altitude residency in summer and low- to mid-altitude residency in winter, a pattern similar to that observed today in the same region [42,44,77]. Moreover, our results also show that the seven horses deposited at the Burgast khirgisuur display distinct mobility patterns. Four of the seven horses (63, 65A and B, 66) exhibit an identical mobility pattern, while the three remaining animals display a range of different patterns. In this discussion, we will first focus on the insights that we can gain from the shared pattern observed in the four horses. We will then discuss the implications of the differences observed between some animals.

### 4.1 Bioavailable isoscape and geographic assignments

To investigate the mobility of Late Bronze Age horses we generated a bioavailable $^{87}Sr/^{86}Sr$ isoscape for the Altai Mountains range but also at a larger scale. Our isoscape (Fig 3) showed consistency with a previous Altai isoscape based on the same plant dataset [43] but allows for spatial assignment at larger scales as random forest predictions expand beyond the area cover by the sampling campaign. Variables retained to train the model and model performance were similar to the original global isoscape [33]. This isoscape represents a first attempt to model the spatial distribution of $^{87}Sr/^{86}Sr$ across Mongolia and is a powerful tool for geographic assignments, allowing going beyond the simple identification of local and non-local individuals by highlighting areas of likely origin [33,34,67].

We used a bayesian assignment approach to take into account the relatively high spatial uncertainty associated with the RF prediction. The Bayesian assignment explicitly integrates uncertainty by identifying the most probable areas based on probability distributions rather than absolute values [67]. Even with high uncertainty, geographic signals remain detectable. Assignment areas are broadened but the relative probability ranking of the most likely regions will remain unchanged. We also aimed at using the relationship between $\delta^{13}C$ and elevation, previously observed in caprines and horses from the Altai region [43] to refine geographic assignments. However, the conversion of intra-tooth values from our individuals did not consistently align with the Altai elevation range, suggesting elevation is not the sole factor driving $\delta^{13}C$ variations recorded in the teeth. Fodder can potentially weaken this relationship. Our experience shows that in modern practices, males are only (lightly) supplemented during harsh years, while females and foals only receive fodder in winter for a limited period of time [43]. Although effect of foddering cannot be entirely ruled out, it is likely marginal, as short-term supplementation in $C_4$ plants should not cause a significant increase in $\delta^{13}C$ values [78].

## 4.2. Cyclical pattern of pastoral mobility during the Late Bronze Age

Strontium isotope analysis of tooth enamel of Bronze Age horses from ST63, 65A-B and 66 indicates that these animals made at least four or five movements per year. However, it is possible that the movements were more frequent than the data set allows us to discern, with some of them not being recorded. This could be due to the animals moving through an isotopically homogeneous zone, or the fact that the residence times were not long enough to allow signal recording. Our previous work, which combined GPS tracking of herds, interviews with herders, and geochemical analysis of horse tooth enamel, demonstrated that residences of less than a month were not discernible in isotopic analyses [38]. In Fig S9 in S1 File we provide the predictions of a mobility model based on GPS data from a modern horse living near Burgast coupled with the regional isoscape. The model predictions indicate that the successive movements of the horse during the year between the plain, the valleys and the alpine plateau lead to an isotope pattern similar to that measured along the tooth enamel of the four ancient horses from the same region. Our modeling approach is validated by the Sr isotope analysis of the tooth of another modern horse from the same herd, which displays the expected isotope pattern.

The similarity between the pattern observed in the modern and the four ancient horses may have its origin in the similarity in climatic and environmental conditions, which, in this part of Mongolia might constrain mobility patterns more than in desert or steppe environments: green areas around rivers from spring on; massive mosquito infestation from summer on, forcing the herders to reach elevated pastures spared from insect invasions; harsh winters with freezing wind corridors in the valleys, while high pastures are inaccessible due to the cold and snow coverage. Paleoclimatic reconstruction based on lake sediment proxies and biomarker compound-specific $\delta^2H$ analyses indicate that warm and wet conditions prevailed in the Mongolian Altai from ~3.5–2.8 to ~2.3–1.5 cal. ka BP, possibly favoring the expansion of mobile nomadic pastoralism in the region [79]. Repetition of the same isotope patterns in M2 and M3 from horses found in ST63 and ST65A indicates that these animals returned to the same pastures from one year to the next. This evidence supports the hypothesis that cyclical pastoral mobility existed as early as the Bronze Age. It is challenging to determine whether this type of mobility pattern is a Bronze Age innovation, as there is a paucity of knowledge regarding the sub-sistence patterns of Neolithic and Eneolithic populations. This is due to the limited availability of archaeological sites that yield faunal remains and other evidence that can be used to ascertain this information. While it is possible to hypothesize a resemblance in mobility patterns between the Bronze Age and the modern era, it is also evident that there is a dearth of data between the Bronze Age and the 21th century. This gap in the historical record must be addressed if we are to gain a deeper understanding of the evolution of human mobility.

Although it is a truism to say that the domestication of the horse enabled long-distance travel, our data indicate that the distances traveled during the Late Bronze Age are relatively modest and likely to be comparable to those in the present era. A modern survey revealed that the cumulated distance traveled each year between the different camps by herders living near Burgast ranges from about 30–270 km [42]. The maximum distance between two consecutive camps is about 30

km and GPS data show that horses can graze on pastures that are up to 45 km apart from each other [38,42]. This does not negate the possibility of long-distance journeys, but our data do not substantiate this hypothesis and suggest that mobility was confined to a rather local level. It is important to note that the first four or five years of an animal's life are the only ones for which we have documented information, as the rest is not recorded in tooth enamel, and therefore remains unknown.

The late Bronze Age period is characterized by the adoption and extensive integration of the horse into economic and cultural spheres. Archaeology shows that multispecies herding was practiced at the time, as remains of horse, cattle, sheep and goat are found in domestic and ritual contexts of the Late Bronze Age of Mongolia [80]. While horses enabled people to move faster and over longer distances, herders were constrained by the speed and mobility of ruminant herds (sheep, goats, cattle), as well as by the weight and volume of the equipment to be conveyed. The other significant advantage is that it permitted more frequent and regular movements, as the limitations and costs associated with energy expenditure and the management and control of the moving herd were reduced. The similarity in pastoral management systems between the Late Bronze Age and the present, characterized by the existence of at least four annual movements, has further implications. It seems plausible to suggest that this may have been linked to the existence of large herds, possibly approaching the size of those that are known to exist today, that is several hundreds. It is reasonable to assume that, above a certain number of individuals, mobility becomes a necessity to avoid depleting food resources, which were certainly richer and more diverse in the LBA due to less impoverished soils and more extensive forests than today. This high degree of residential mobility, which existed 3,000 years ago, is thus associated with the enhanced herding capacity made possible by the horse. All in all, our data support the hypothesis that pastoral practices were adapted to a territory with a potentially large animal load.

Given the similarity in their movement profiles, one might wonder if these four horses did not belong to the same herd. Firstly, even if the radiocarbon dates are not significantly different, the degree of precision of the calibrated ranges (typically one or two centuries) is insufficient to demonstrate that the animal lived at the same time. Secondly, it is important to keep in mind that the horses deposited at the khirgisuur exhibited varying ages at death, with a range of about 1–20 years. Although it can be reasonably inferred that horse slaughter occurred at the onset of winter, as is currently the case, previous work by our team showed that not all horses in a khirgisuur died at the same time of the year [14]. Moreover, some skulls were not deposited fresh around the khirgisuur, opening the possibility that the deposits are distributed over time. This complicates the reconstruction of temporal relationships between the various deposits. If the pastoralists' annual route remains consistent from one year to the next, all of their horses will exhibit a similar isotopic signal. It is therefore more accurate to suggest that the four horses likely belonged to the same family (who can keep the same routes over several generations) rather than that they came from the same herd. It would be tempting here to consider that we have characterized the existence of a herd for the first time, using isotopic tools, but the archaeological and cultural context reveals a complexity that goes beyond first impressions.

### 4.3. Inter-individual variability in horse mobility patterns

While four of the seven horses (63, 65A and B, 66) exhibit an identical mobility pattern, the remaining three display a range of different patterns. Horse 61 is too young to exhibit a complete yearly cycle; nevertheless, its Sr value is consistent with that expected from a local origin. It is possible that the pattern observed in this horse is analogous to that observed in the other four. However, it is challenging to assess this with certainty. Horse from ST67 has a completely flat Sr pattern in its $M_1$, $M_2$ and $M_3$. The flat profile may be attributed to either the absence of movement by the horse during its first 5–6 years of life, or a permanent foddering of the animal, which could have served as riding animal; alternatively, it could be the result of the animal's movements between areas with homogeneous isotopic composition. Finally, one horse (ST62) was not native of the area of Burgast, as suggested by the low Sr values measured in the upper part of its first molar. Modeling suggests that he was born at a minimum distance of 50 km from the area of Burgast, then brought

during its 3rd year of life near Burgast where it died. It is difficult to establish where this animal was sourced. The areas in closest proximity are detected approximately 50 km to the north-west (present-day Russian Altai) and south/south-west (present-day Mongolian Altai) but areas detected away from the Altai are also possible. However, these areas must be treated with caution as the low $^{87}Sr/^{86}Sr$ values of ST62 fall out of the isoscape range, meaning the assignment of these samples may be biased toward high-uncertainty areas where extreme values are more probable. Despite the highly predictive power of the random forest regression, bioavailable $^{87}Sr/^{86}Sr$ prediction in areas with very low or no sampling effort are weaker [33,36] and would require additional samples across Mongolia to refine the isoscape. It bears noting that the Altai Mountains and their neighboring ranges are home to some of the largest mineral deposits in this part of Asia. The extraction of raw materials and their trade in the form of raw or processed products (e.g., [81,82]) are the primary drivers of the dynamism observed in this region, with large-scale exchange networks and cultural contacts have occasionally/potentially spanned great distances (e.g., [83,84]). These far-reaching interactions can be traced back to the beginning of the 2nd millennium BC. It is thus unsurprising that animals, particularly horses, reflect these interactions by the end of the 2nd millennium BCE, becoming a key component of such exchanges.

In any case, the data indicate that six of the seven horses most likely lived in the vicinity of Burgast. The question thus arises as to whether the seventh horse belonged to the same family that changed its grazing area over time. The discrepancies in equine mobility patterns can be elucidated in two ways. Firstly, these differences can be seen as an illustration of the adaptability of nomadic pastoralism, provided that the herd in question is considered as a single entity. The flexibility of contemporary nomadic lifestyles allows communities to respond to changing conditions, which particularly allows them to mitigate risks associated with climate variability for example. While the nomadic itineraries of herders are typically cyclical, variations in seasonal mobility practices, whether for socio-economic or environmental reasons, are thus common and have also been locally documented in Mongolia, including on an interannual scale (e.g., in Bayan-Ölgii Province [42,77,85], and in Arkhangai Province [76,86,87]).

The dissimilarities in mobility patterns may also be attributed to a transfer of individuals between herds, thereby illustrating the interconnectivity between herding communities. According to this hypothesis, the horse from ST62 could have been obtained by the family through acquaintances. Animal circulation between herders, which can take the form of exchange, trade, gift or theft, are well documented in social anthropology (e.g., [88,89] for African pastoralists). Although anthropological study can help us keeping in mind the variety of possible circulation modalities, archaeology can, however, not document the intricacies of exchange practices for the contexts of late Bronze Age Mongolia. Additionally, it is important to acknowledge the potential for bias in the selection of individuals studied, as they were possibly chosen based on certain criteria, and may not be a representative sample of the larger horse population. Nevertheless, this result illustrates the connectivity, direct or indirect, between territories located at a relatively large distance (50–100 km) from the Burgast khirgisuur. This study should be extended to encompass horses from other khirgisuur herds in order to gain a comprehensive understanding of the extent of local variation.

### 4.4. Horse deposits and rituals

In addition to shedding light on the study of herd management and movement, the data provide new insights into ritual practices. Indeed, among the questions relating to the meaning, implementation, and operation of the funerary complexes, that of the origin of the horses deposited is of particular significance. The smaller khirgisuur are typically interpreted as reflecting family-type installations, while the larger ones must have been linked to larger communities expanding beyond the closest family members [90,91]. Until this study, however, there was no evidence to support this hypothesis, and other explanations could be proposed. For example, the horses found in the Burgast structure could have originated from different families, residing in disparate locations, and may have been presented as gifts or tribute marks. Our findings indicate that the majority of these horses belong to the same family, or at least families with the same mobility patterns. Consequently, they are likely to have originated from the deceased's inner circle. In the future, it would prove beneficial to apply

this approach in a systematic fashion to the larger khirgisuurs in order to ascertain the total number of participating families, or at least groups sharing a similar mobility pattern, involved in the deposits. Notwithstanding the numerous uncertainties that persist, our findings contribute to the growing body of knowledge regarding the ritual landscape of Bronze Age communities.

## Supporting information

**S1 File. Supplementary material SM1 to SM6 with Fig S1-S9 and Table S1-S7.**
(DOCX)

**S2 File. Supplementary material SM7_Table S8. Global bioavailable $^{87}$Sr/$^{86}$Sr database.**
(XLSX)

**S3 File. Supplementary material SM8_Table S9. Isotope data ($\delta^{18}$O, $\delta^{13}$C, $^{87}$Sr/$^{86}$Sr) measured on the 7 Bronze Age horses and the modern horse.**
(XLSX)

**S4 File. Supplementary material SM9. Archive containing videos of geographic assignments for the 7 Bronze Age horses and the modern horse.**
(RAR)

**S5 File. Supplementary material SM10. Isoscape and geographic assignment R scripts.**
(DOCX)

**S6 File. Supplementary material SM11. Archive containing the strontium isoscape.**
(RAR)

## Acknowledgments

The isotopic analysis of this study was made possible by the INSU/CNRS MC-ICP-MS national facility at ENS-Lyon. We thank the staff at Elemental Scientific Lasers for providing a laser and technical support as part of the formal collaboration with ENS-Lyon.

## Author contributions

**Conceptualization:** Antoine Zazzo, Aurélie Coulon, Sébastien Lepetz.

**Data curation:** Antoine Zazzo, Maël Le Corre, Nicolas Lazzerini.

**Formal analysis:** Antoine Zazzo, Maël Le Corre, Nicolas Lazzerini, Philippe Telouk, François Thil.

**Funding acquisition:** Antoine Zazzo, Aurélie Coulon, Sébastien Lepetz.

**Investigation:** Antoine Zazzo, Nicolas Lazzerini, Charlotte Marchina, Noost Bayarkhuu, Vincent Bernard, Mathilde Cervel, Denis Fiorillo, Dominique Joly, Michel Lemoine, Philippe Telouk, François Thil, Tsagaan Turbat, Vincent Balter, Sébastien Lepetz.

**Methodology:** Antoine Zazzo.

**Project administration:** Antoine Zazzo, Aurélie Coulon, Sébastien Lepetz.

**Resources:** Antoine Zazzo, Noost Bayarkhuu, Vincent Bernard, Mathilde Cervel, Denis Fiorillo, Dominique Joly, Michel Lemoine, François Thil, Tsagaan Turbat, Vincent Balter, Sébastien Lepetz.

**Supervision:** Antoine Zazzo, Vincent Balter, Aurélie Coulon, Sébastien Lepetz.

**Validation:** Antoine Zazzo.

**Visualization:** Antoine Zazzo, Sébastien Lepetz.

**Writing – original draft:** Antoine Zazzo, Maël Le Corre, Sébastien Lepetz.

**Writing – review & editing:** Antoine Zazzo, Maël Le Corre, Nicolas Lazzerini, Charlotte Marchina, Noost Bayarkhuu, Vincent Bernard, Mathilde Cervel, Tsagaan Turbat, Sébastien Lepetz.

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
