## [Decision Letter · Decision Letter 0]

4 Feb 2025

PONE-D-24-560433000 yr-old patterns of mobile pastoralism revealed by multiple isotopes and radiocarbon dating of ancient horses from the Mongolian Altai.PLOS ONE

Dear Dr. Zazzo,

Thank you for submitting your manuscript to PLOS ONE. After careful consideration, we feel that it has merit but does not fully meet PLOS ONE’s publication criteria as it currently stands. Therefore, we invite you to submit a revised version of the manuscript that addresses the points raised during the review process.

The manuscript is overall well-written and the work of interest. I concur with reviewers that you should propose alternative explanations for the non-local signatures of some of the horses and/or in general tone down your interpretations.

Here a few minor edits from myself:

Line 261: please use laser energy (fluence J/cm^2) rather than a %.

Line 267: the mean value of nist1400 has 5 decimals while the error has 6.

Line 286: As disclosed by Bataille and coauthors in their own 2020 paper on the global isoscape, using the whole global dataset for predicting an under-represented area can decrease the accuracy of the prediction itself (see the Madagascar example). I’m not saying you should repeat the whole analysis, yet you should likely test the performance of the RF model with the Altai data only.

Line 422: This sentence can be interpreted as environmental changes of the Sr (as observed e.g. in tropical areas) rather than a seasonal mobility of the horse.

We look forward to receiving your revised manuscript.

Kind regards,

Federico Lugli, Ph.D.

Academic Editor

PLOS ONE

Journal Requirements:

Please ensure that your manuscript meets PLOS ONE's style requirements, including those for file naming. The PLOS ONE style templates can be found at https://journals.plos.org/plosone/s/file?id=wjVg/PLOSOne_formatting_sample_main_body.pdf and https://journals.plos.org/plosone/s/file?id=ba62/PLOSOne_formatting_sample_title_authors_affiliations.pdf 2. In your manuscript, please provide additional information regarding the specimens used in your study. Ensure that you have reported human remain specimen numbers and complete repository information, including museum name and geographic location.  If permits were required, please ensure that you have provided details for all permits that were obtained, including the full name of the issuing authority, and add the following statement: 'All necessary permits were obtained for the described study, which complied with all relevant regulations.' If no permits were required, please include the following statement: 'No permits were required for the described study, which complied with all relevant regulations.' For more information on PLOS ONE's requirements for paleontology and archeology research, see https://journals.plos.org/plosone/s/submission-guidelines#loc-paleontology-and-archaeology-research. 3. Please include a complete copy of PLOS’ questionnaire on inclusivity in global research in your revised manuscript. Our policy for research in this area aims to improve transparency in the reporting of research performed outside of researchers’ own country or community. The policy applies to researchers who have travelled to a different country to conduct research, research with Indigenous populations or their lands, and research on cultural artefacts. The questionnaire can also be requested at the journal’s discretion for any other submissions, even if these conditions are not met.  Please find more information on the policy and a link to download a blank copy of the questionnaire here: https://journals.plos.org/plosone/s/best-practices-in-research-reporting. Please upload a completed version of your questionnaire as Supporting Information when you resubmit your manuscript. 4. Please note that PLOS ONE has specific guidelines on code sharing for submissions in which author-generated code underpins the findings in the manuscript. In these cases, we expect all author-generated code to be made available without restrictions upon publication of the work. Please review our guidelines at https://journals.plos.org/plosone/s/materials-and-software-sharing#loc-sharing-code and ensure that your code is shared in a way that follows best practice and facilitates reproducibility and reuse. 5. Thank you for stating in your Funding Statement: Fieldwork was supported by the Joint French-Mongol Archaeological Mission (Ministère des Affaires Etrangères et du Développement International, director S. Lepetz) and by the CNRS. This research was funded by a Ph.D. grant to NL from the French National Research Agency/Agence Nationale de la Recherche Laboratoires d'Excellence ANR-10-LABX-0003-BCDiv, in the context of the “Investissements d’avenir” ANR-11-IDEX-0004-02, by a postdoctoral fellowship to MLC from the French National Research Agency/Agence Nationale de la Recherche ANR-20-CE27-0018 and by an Inalco (Institut National des Langues et Civilisations Orientales) Early Career Research Grant 2017 to CM Please provide an amended statement that declares *all* the funding or sources of support (whether external or internal to your organization) received during this study, as detailed online in our guide for authors at http://journals.plos.org/plosone/s/submit-now.  Please also include the statement “There was no additional external funding received for this study.” in your updated Funding Statement. Please include your amended Funding Statement within your cover letter. We will change the online submission form on your behalf. 6. Thank you for stating the following in the Acknowledgments Section of your manuscript: Fieldwork was supported by the Joint French-Mongol Archaeological Mission (Ministère des Affaires Etrangères et du Développement International, director S. Lepetz) and by the CNRS. This research was funded by a Ph.D. grant to NL from the French National Research Agency/Agence Nationale de la Recherche Laboratoires d'Excellence ANR-10-LABX-0003-BCDiv, in the context of the “Investissements d’avenir” ANR-11-IDEX-0004-02, by a postdoctoral fellowship to MLC from the French National Research Agency/Agence Nationale de la Recherche ANR-20-CE27-0018 and by an Inalco (Institut National des Langues et Civilisations Orientales) Early Career Research Grant 2017 to CM. We note that you have provided funding information that is not currently declared in your Funding Statement. However, funding information should not appear in the Acknowledgments section or other areas of your manuscript. We will only publish funding information present in the Funding Statement section of the online submission form. Please remove any funding-related text from the manuscript and let us know how you would like to update your Funding Statement. Currently, your Funding Statement reads as follows: Fieldwork was supported by the Joint French-Mongol Archaeological Mission (Ministère des Affaires Etrangères et du Développement International, director S. Lepetz) and by the CNRS. This research was funded by a Ph.D. grant to NL from the French National Research Agency/Agence Nationale de la Recherche Laboratoires d'Excellence ANR-10-LABX-0003-BCDiv, in the context of the “Investissements d’avenir” ANR-11-IDEX-0004-02, by a postdoctoral fellowship to MLC from the French National Research Agency/Agence Nationale de la Recherche ANR-20-CE27-0018 and by an Inalco (Institut National des Langues et Civilisations Orientales) Early Career Research Grant 2017 to CM Please include your amended statements within your cover letter; we will change the online submission form on your behalf. 7. We note that Figures 1, 3, 6B, S6, S7, S8, S9, S10 and S11 in your submission contain map images which may be copyrighted. All PLOS content is published under the Creative Commons Attribution License (CC BY 4.0), which means that the manuscript, images, and Supporting Information files will be freely available online, and any third party is permitted to access, download, copy, distribute, and use these materials in any way, even commercially, with proper attribution. For these reasons, we cannot publish previously copyrighted maps or satellite images created using proprietary data, such as Google software (Google Maps, Street View, and Earth). For more information, see our copyright guidelines: http://journals.plos.org/plosone/s/licenses-and-copyright. We require you to either present written permission from the copyright holder to publish these figures specifically under the CC BY 4.0 license, or remove the figures from your submission: a. You may seek permission from the original copyright holder of Figures 1, 3, 6B, S6, S7, S8, S9, S10 and S11 to publish the content specifically under the CC BY 4.0 license.   We recommend that you contact the original copyright holder with the Content Permission Form (http://journals.plos.org/plosone/s/file?id=7c09/content-permission-form.pdf) and the following text:“I request permission for the open-access journal PLOS ONE to publish XXX under the Creative Commons Attribution License (CCAL) CC BY 4.0 (http://creativecommons.org/licenses/by/4.0/). Please be aware that this license allows unrestricted use and distribution, even commercially, by third parties. Please reply and provide explicit written permission to publish XXX under a CC BY license and complete the attached form.” Please upload the completed Content Permission Form or other proof of granted permissions as an "Other" file with your submission. In the figure caption of the copyrighted figure, please include the following text: “Reprinted from [ref] under a CC BY license, with permission from [name of publisher], original copyright [original copyright year].” b. If you are unable to obtain permission from the original copyright holder to publish these figures under the CC BY 4.0 license or if the copyright holder’s requirements are incompatible with the CC BY 4.0 license, please either i) remove the figure or ii) supply a replacement figure that complies with the CC BY 4.0 license. Please check copyright information on all replacement figures and update the figure caption with source information. If applicable, please specify in the figure caption text when a figure is similar but not identical to the original image and is therefore for illustrative purposes only.The following resources for replacing copyrighted map figures may be helpful: USGS National Map Viewer (public domain): http://viewer.nationalmap.gov/viewer/The Gateway to Astronaut Photography of Earth (public domain): http://eol.jsc.nasa.gov/sseop/clickmap/Maps at the CIA (public domain): https://www.cia.gov/library/publications/the-world-factbook/index.html and https://www.cia.gov/library/publications/cia-maps-publications/index.htmlNASA Earth Observatory (public domain): http://earthobservatory.nasa.gov/Landsat:
http://landsat.visibleearth.nasa.gov/USGS EROS (Earth Resources Observatory and Science (EROS) Center) (public domain): http://eros.usgs.gov/#Natural Earth (public domain): http://www.naturalearthdata.com/ 8. We note that Figure S1 in your submission contain copyrighted images. All PLOS content is published under the Creative Commons Attribution License (CC BY 4.0), which means that the manuscript, images, and Supporting Information files will be freely available online, and any third party is permitted to access, download, copy, distribute, and use these materials in any way, even commercially, with proper attribution. For more information, see our copyright guidelines: http://journals.plos.org/plosone/s/licenses-and-copyright. We require you to either present written permission from the copyright holder to publish these figures specifically under the CC BY 4.0 license, or remove the figures from your submission: a. You may seek permission from the original copyright holder of Figure S1 to publish the content specifically under the CC BY 4.0 license.  We recommend that you contact the original copyright holder with the Content Permission Form (http://journals.plos.org/plosone/s/file?id=7c09/content-permission-form.pdf) and the following text:“I request permission for the open-access journal PLOS ONE to publish XXX under the Creative Commons Attribution License (CCAL) CC BY 4.0 (http://creativecommons.org/licenses/by/4.0/). Please be aware that this license allows unrestricted use and distribution, even commercially, by third parties. Please reply and provide explicit written permission to publish XXX under a CC BY license and complete the attached form.” Please upload the completed Content Permission Form or other proof of granted permissions as an "Other" file with your submission.  In the figure caption of the copyrighted figure, please include the following text: “Reprinted from [ref] under a CC BY license, with permission from [name of publisher], original copyright [original copyright year].” b. If you are unable to obtain permission from the original copyright holder to publish these figures under the CC BY 4.0 license or if the copyright holder’s requirements are incompatible with the CC BY 4.0 license, please either i) remove the figure or ii) supply a replacement figure that complies with the CC BY 4.0 license. Please check copyright information on all replacement figures and update the figure caption with source information. If applicable, please specify in the figure caption text when a figure is similar but not identical to the original image and is therefore for illustrative purposes only. 9. We are unable to open your Supporting Information file "SM10_script_isoscape_and_assignment.R". Please kindly revise as necessary and re-upload. 10. Please review your reference list to ensure that it is complete and correct. If you have cited papers that have been retracted, please include the rationale for doing so in the manuscript text, or remove these references and replace them with relevant current references. Any changes to the reference list should be mentioned in the rebuttal letter that accompanies your revised manuscript. If you need to cite a retracted article, indicate the article’s retracted status in the References list and also include a citation and full reference for the retraction notice.

Reviewers' comments:

Reviewer's Responses to Questions

Comments to the Author

1. Is the manuscript technically sound, and do the data support the conclusions?

Reviewer #1: Yes

Reviewer #2: Yes

2. Has the statistical analysis been performed appropriately and rigorously?

Reviewer #1: Yes

Reviewer #2: Yes

3. Have the authors made all data underlying the findings in their manuscript fully available?

Reviewer #1: Yes

Reviewer #2: Yes

4. Is the manuscript presented in an intelligible fashion and written in standard English?

Reviewer #1: Yes

Reviewer #2: No

5. Review Comments to the Author

**Reviewer #1:**  I believe that Zazzo et al.’s paper represents a good example of multi-isotope analysis for exploring animal mobility in the context of Western Mongolia and a valuable contribution for mobility studies in the area, as it presents an updated regional isoscape of bioavailable Sr. The methodology is sound (I very much appreciated the use of modern horse as a check-test), results are consistent, and references updated. Therefore, I recommend the publication of the manuscript on PLOS ONE, although I suggest to address some relevant issues regarding the general structure of the paper and the final interpretation of the isotopic results, as regards to the relationship with the archaeological context.

First, to improve the clarity of the main text, I think that there are too many references to the supplementary information. As PLOS ONE has no page limit, perhaps some of the figures that are currently in the SI, might be put in the main text. FigS1, FigS2, for example, are extremely important for the interpretation of the context.

I found the presentation of the isotopic results a bit hard to read. Instead of subdividing paragraphs by isotopes, why not by horse? As only 6 horses are analysed in such a detailed resolution, it would be great to present each single “life history”, and only at the end, a synthesis of all data.

My second point concerns the archaeological contexts. Can we exclude that the horses were sacrificed for celebrating the deceased’s death? I read at lines 602-603 that “previous work by our team showed that not all horses in a khirgisuur died at the same time of the year”. Looking at Lazzerini et al.’s paper (2020), only two horses were analysed, ST 61 and 66. The skulls ST63, 64, 65, 66 might therefore be simultaneous (also judging from the planimetry). The pits in which skulls were deposed, in fact, seem well connected to the burial (FigS1). In this case, a single event, at least for ST63, 64, 65, 66 cannot be ruled out, as far as I understood. This would be corroborated by the homogeneity of radiocarbon dates (0-104 y, see line 356), with the obvious limitations derived from the 14C resolution. Eventually stratigraphic evidence can help in addressing the issue of synchrony /diachrony. For example, ST63, 64, 65, 66 structures are very close to each other. Do some of the pits cut other pits (which would indicate a sequence of events delayed in time)? Moreover, the deposition of animal skulls, at least to my opinion, seems more typical of ritual performed in funerary areas, as documented in various Bronze and Iron Age contexts across Eurasia.

At line 157-161 the Authors state that “On the eastern side, there are seven mounds (ST 61 to 67) yielding horse remains of 7 individuals, mainly represented by their skulls or mandibles and sometimes accompanied by connected cervical vertebrae and terminal phalanges. The horse remains are fairly well preserved, but bone displacements, breakage and missing parts seem to indicate that taphonomic disturbances (possibly by burrowing animals) have occurred, although it is not always possible to precisely define their extent”. As expert of human and animal bones, I can say that it is quite a unique case in which cervical vertebrae and phalanges are preserved, and long bones diaphysis are not. Trabecular bone is much more fragile than cortical bone. Therefore, isn’t it possible that the skulls (+ cervical vertebrae, etc.) were the only parts that were deposed? And why only cervical vertebrae and not thoracic/lumbar/sacral/caudal vertebrae? Is it a coincidence, or the horses were beheaded and sacrificed? Moreover, what is the size of the deposition pits (it is not very clear from Fig1)? Are they large enough to contain the whole body of a horse (they are not, judging from Lazzerini et al. 2020, Fig. 2, which would be much better than Fig.1 of the manuscript)? This is just a hypothesis but, looking at the way the evidence is presented in the manuscript, the most convincing, at least for individuals 63-66. If the authors have archaeological or archaeozoological data to exclude it, I suggest addressing the issue in the main text.

Following this line, in case the horses were sacrificed, for example, should we consider other alternative ideas, for example that the horses were spoils of war (taken from different enemies)? Celebration of victory in conflict? Gifts to a war chief? Sacrificial victims for accessing the afterlife? In that case, their mobility (or the mobility of some of them) could be related to different purposes that simply grazing. What I am emphasizing, for the benefit of the authors, is that the thesis they present assumes that the deposition mirrors simply herds, or different herds. I do not disagree. However, the presentation of the archaeological context should be more detailed, and the final conclusion should, at least in my opinion, taking into account different (and more complex) alternative options.

Minor comments:

I see a slight, partial contradiction between the straightforward statement at lines 67-69 “Mounted nomadic pastoralism was adopted in Mongolia at the beginning of the Late Bronze Age (c. 1200 BCE) and played a significant role in shaping the social, economic and cultural landscape of the region” and the statement in the conclusion at lines 656-658 “Although anthropological study can help us keeping in mind the variety of possible circulation modalities, archaeology can, however, not document the intricacies of exchange practices for the contexts of late Bronze Age Mongolia”. Either we have a clear picture of LBA pastoralism in Mongolia, or archaeological evidence of animal circulation is too scarce/not deeply investigated. Eventually instead of just mentioning Honeychurch’s works, it would be clearer for the readers what the LBA archaeological evidence tell us about this significant role.

647: in which sense “contemporary nomadic lifestyles […] reflect responses to socio-economic conditions”? Does it mean that herder change routes depending on the economic resources they have or that herders having different economic statuses use different routes/pastures?

658: Late Bronze Age Mongolia

**Reviewer #2: ** The article by Zazzo et al regarding mobile pastoralism in Mongolia is very well done on the scientific side. Their use of combined Sr, O, and C isotopes is well thought out and the results are promising. I agree with the majority of their findings, but they need to be stated more strongly in the discussion and conclusion. About midway through the paper the text becomes a bit repetitive and needs some editing. It might be that some of the detail of isoscape construction, etc, could be moved to the supplement.

One issue with the paper is the consistent use of the term nomadic, even when they are proposing to study mobility. This is seen in the introduction, line 66 when they discuss the adoption of ‘mounted nomadic pastoralism’ being adopted. It remains unclear why they did not mention mounted pastoralism, as the ‘nomadic’ part is the question under study. Were early horse riders nomadic, or not?

This happens again on line 90, where they say that they want to address the study of pastoral nomadic mobility – it is enough to say pastoral mobility. The word nomadic implies that you already know the answer to the question.

Starting at line 211, there are some issues in this section. Line 217, ‘changes in the relative abundance of WILD C3 and C4 plants’ – but there should be a caveat here that the carbon isotope values of enamel could also support evidence for foddering with wild or domesticated plants – which is part of the reason the carbon isotopes of enamel cannot be used as a proxy for elevation – because humans are managing horses.

I am not sure that the Bayesian geographic assignment was necessary, but if you are going to use this then please alter the figures so that it is clear that the colors are elevation rather than isotope values. It is pretty confusing.

Lines 393 through 404, this description of correlations between carbon isotopes of tooth enamel and elevation is spurious. It does not help with your argument and actually incorporates confusion into your argument. I would cut this and any discussion of possible correlations between d13C of enamel and predictions of elevation.

Many of the figures, especially 4 and 5, are very blurry and unreadable. Please remake or increase the size.

The geographic assignments are ok, but the maps are unclear. The yellow and red (supposed to be elevation?) do not help the reader. It would be better if these were given colors that correspond to strontium isotope values so that we can see the various locations where horses may have spent time.

Smaller detailed edits:

Line 76 should read ‘Some khirigsuurs are quite small containing fewer than ten horse head mounds, while others can reach hundreds of horse head mounds’

Line 153 should say ‘…with most khirigsuurs, the main burial is associated….’

Line 345 the word Given should be lowercase

Lines 368 to 369, the authors need to explain the RMSE and why they think the Rsquared value of 0.67 is important.

Line 390, what is a solution analysis in this context?

Line 392, this should say ‘…and predicted altitude’

Line 500 should say ‘…historical pastoral systems’

Line 506 remove nomadic from these discussions, it should say mobility practices. Mountain nomadism should be vertical mobility

Lines 519 to 523 – it seems like the spatial error is significant. How can spatial error of >0.001 not be problematic?

Line 603 – remove ‘his horses’, maybe their horses?

Lines 644, 645 – remove nomadic when discussing mobility practices.

6. PLOS authors have the option to publish the peer review history of their article (what does this mean? ). If published, this will include your full peer review and any attached files.

**Do you want your identity to be public for this peer review?** For information about this choice, including consent withdrawal, please see our Privacy Policy .

Reviewer #1: **Yes: ** Claudio Cavazzuti

Reviewer #2: No

---

## [Author Response · Author response to Decision Letter 1]

4 Mar 2025

please see our point by point response to the reviewers and editor comments in the attached document.

---

## [Editor Report · Decision Letter 1]

23 Mar 2025

3000 yr-old patterns of mobile pastoralism revealed by multiple isotopes and radiocarbon dating of ancient horses from the Mongolian Altai.

PONE-D-24-56043R1

Dear Dr. Zazzo,

We’re pleased to inform you that your manuscript has been judged scientifically suitable for publication and will be formally accepted for publication once it meets all outstanding technical requirements.

Kind regards,

Federico Lugli, Ph.D.

Academic Editor

PLOS ONE
---

## [Editor Report · Acceptance letter]

PONE-D-24-56043R1

PLOS ONE

Dear Dr. Zazzo,

I'm pleased to inform you that your manuscript has been deemed suitable for publication in PLOS ONE. Congratulations! Your manuscript is now being handed over to our production team.

Kind regards,

on behalf of

Dr. Federico Lugli

Academic Editor

PLOS ONE